# LOCAL KL CONVERGENCE RATE FOR STEIN VARIATIONAL GRADIENT DESCENT WITH REWEIGHTED KERNEL

## ABSTRACT

We study the convergence properties of Stein Variational Gradient Descent (SVGD) algorithm for sampling from a non-normalized probabilistic distribution $p_*(\boldsymbol{x}) \propto \exp(-f_*(\boldsymbol{x}))$. Compared with Kernelized Stein Discrepancy (KSD) convergence analyzed in previous literature, KL convergence as a more convincing criterion can better explain the effectiveness of SVGD in real-world applications. In the population limit, SVGD performs smoothed gradient descent with kernel integral operator. Notably, SVGD with smoothing kernels suffers from gradient vanishing in low-density areas, which makes the error term between smoothed gradient and the Wasserstein gradient not controllable. In this context, we introduce a reweighted kernel to amplify the smoothed gradient in low-density areas, which leads to a bounded error term. When the $p_*(\boldsymbol{x})$ satisfies log-Sobolev inequality, we develop the convergence rate for SVGD in KL divergence with the reweighted kernel. Our analysis points out the defects of conventional smoothing kernels in SVGD and provides the convergence rate for SVGD in KL divergence.

## 1 INTRODUCTION

Sampling from non-normalized distributions is a crucial task in statistics. In particular, in Bayesian inference, Markov Chain Monte Carlo (MCMC) and Variational Inference (VI) are considered two mainstream lines to handle the intractable integration of posterior distributions. On the one hand, although methods based on MCMC, e.g., Langevin Monte Carlo (Durmus & Moulines, 2019; Welling & Teh, 2011) (LMC) and Metropolis-adjusted Langevin algorithm (Xifara et al., 2014) (MALA), are able to provide approximate target distributions with arbitrarily small error (Wibisono, 2018), the sample efficiency is low due to the lack of repulsive force between samples (Duncan et al., 2019; Korba et al., 2020). On the other hand, VI-based sampling methods (Blei et al., 2017; Ranganath et al., 2014) can improve the sampling efficiency by reformulating inference as an optimization problem. However, restricting the search space of the optimization problem to some parametric distributions in VI usually causes a huge gap between its solution and the target distribution $p_*$.

Inspired by conventional VI, a series of recent works analyze LMC as the optimization problem of Kullback-Leibler (KL) divergence (Wibisono, 2018; Bernton, 2018; Durmus et al., 2019), i.e.,

$$\underset{p \in \mathcal{P}_2(\mathbb{R}^d)}{\arg\min} \; H_{p_*}(p) \coloneqq D_{\mathrm{KL}}(p\|p_*) = \int p(\boldsymbol{x}) \ln \frac{p(\boldsymbol{x})}{p_*(\boldsymbol{x})} d\boldsymbol{x} \tag{1}$$

where $\mathcal{P}_2(\mathbb{R}^d)$ is the set of Radon-Nikodym derivatives of probability measures $\nu$ over Lebesgue measure such that $p(\boldsymbol{x}) = \mathrm{d}\nu(\boldsymbol{x})/\mathrm{d}\boldsymbol{x}$, $\int \|\boldsymbol{x}\|^2 p(\boldsymbol{x})\mathrm{d}\boldsymbol{x} < \infty$. LMC is considered as a discrete scheme of the gradient flow of the relative entropy by driving particles with stochastic and energy-induced force. Besides, to take the best of both MCMC and VI, Stein Variational Gradient Descent (Liu & Wang, 2016) (SVGD) was proposed as a non-parametric VI method. It replaces the stochastic force in LMC with the interaction between particles and approximates the target distribution by a driving force in Reproducing Kernel Hilbert space (RKHS). It means the gradient flow of SVGD is defined by the functional derivative projection of Eq. 1 to RKHS. The empirical performance of SVGD and its variants have been largely demonstrated in various tasks such as learning deep probabilistic models (Liu & Wang, 2016; Pu et al., 2017), Bayesian inference (Liu & Wang, 2016; Feng et al., 2017; Detommaso et al., 2018), and reinforcement learning (Liu et al., 2017).

In addition to rich applications, there is a lot of work on the theoretical analysis of SVGD. For example, Kernelized Stein Discrepancy (KSD) convergence properties of SVGD under asymptotic and non-asymptotic settings are investigated by Liu (2017); Lu et al. (2019) and Korba et al. (2020); Salim et al. (2021; 2022), respectively. However, different from the convergence of KL divergence in the analysis of LMC (Cheng & Bartlett, 2018; Vempala & Wibisono, 2019), KSD convergence cannot deduce the effectiveness of SVGD in some real-world applications, e.g., posterior sampling (Welling & Teh, 2011) and non-convex learning (Raginsky et al., 2017). Then, to provide KL convergence, some other works (Duncan et al., 2019; Korba et al., 2020) present a linear convergence of SVGD with Stein log-Sobolev inequality (SLSI) (Duncan et al., 2019). Nonetheless, different from the clear meaning and criteria of standard log-Sobolev inequality (LSI) in the analysis of LMC (Vempala & Wibisono, 2019), the establishment of SLSI requires the property of the coupling of designed smoothing kernels and the target distribution, which can hardly be verified in commonly used kernels (Duncan et al., 2019). In addition, SLSI in higher dimensions is more challenging to hold.

To fill these gaps, in this paper, we aim to provide the convergence rate of SVGD (in the infinite particle regime) in terms of KL objective, when $p_* = e^{-f_*}$ satisfies standard LSI. Specifically, we first point out that the SVGD with smoothing kernel, e.g., RBF kernel, suffers from gradient vanishing in low-density areas due to the extra $p_t(\boldsymbol{x})$ scaling. Then, we denote the importance of reweighted kernels by dividing $p_t(\boldsymbol{x})$ or $p_*(\boldsymbol{x})$, where the scaling of smoothed gradients can be normalized. With the reweighting scaling $p_*^{-1/2}(\boldsymbol{x})p_*^{-1/2}(\boldsymbol{y})$ for kernel $k(\boldsymbol{x}, \boldsymbol{y})$ and regularity conditions, SLSI in higher dimensions can be nearly established with an additional term controlled by kernel approximation error. Finally, by choosing a proper reweighted smoothing kernel, the KL divergence of SVGD dynamics obtains a local linear convergence rate to any neighborhood of $p_*(\boldsymbol{x})$ under mild assumptions when the initialization $p_0(\boldsymbol{x})$ is relatively close to $p_*(\boldsymbol{x})$.

The main contributions of the paper are as follows:

- We introduce reweighted kernels to SVGD which replaces traditional smoothing kernels and overcomes the gradient vanishing problem in low-density areas.
- We study the KL convergence rate of SVGD algorithm. Under the standard LSI and some mild assumptions, we show SVGD with a reweighted kernel has a local linear convergence rate to any neighborhood of $p_*(\boldsymbol{x})$.

## 2 PRELIMINARIES

In this section, we first introduce important notations used in the following sections. Then, we explain how to optimize functionals on Wasserstein space by continuous updates in the infinite particle regime. After that, we show that the key condition LSI on the target distribution to obtain the KL convergence rate of LMC. However, the convergence rate of SVGD dynamics is non-trivial with this assumption.

**Notations.** In following sections, bold letters $\boldsymbol{x}, \boldsymbol{y}, \boldsymbol{z}$ denote vectors in $\mathbb{R}^d$, and $\mathcal{B}(\boldsymbol{x}, r)$ means the open ball centered at $\boldsymbol{x}$ with radius $r > 0$. For function $f : \mathbb{R}^d \to \mathbb{R}$, $\nabla f(\cdot)$ and $\nabla^2 f(\cdot)$ refer to its gradient and Hessian matrix respectively. For function $f : \mathbb{R}^d \to \mathbb{R}^d$, $\nabla f(\cdot)$ and $\nabla \cdot f(\cdot)$ present the Jacobian matrix and divergence. For function with multiple variables, $\nabla_i$ means the gradient w.r.t $i$-th variable. The distributions are assumed to be absolutely continuous with respect to the Lebesgue measure, which produces density function $p$. The probability density function of the target posterior is denoted by $p_*$. The density at time $t$ is $p_t$. Notation $\| \cdot \|$ denotes 2-norm for both vectors and matrix. In Hilbert space $\mathcal{H}$ equipped with the inner product $\langle \cdot, \cdot \rangle_{\mathcal{H}}$, the norm is induced as $\| \cdot \|_{\mathcal{H}}$. The set $\mathcal{P}_2(\mathcal{X})$ is consist of probability measure $\mu$ on $\mathcal{X}$ with finite second order moment. $\tilde{k}$ denotes a function smoother, such as $\exp(-\|x\|^2)$, $\max\{0, 1 - \|x\|\}$.

### 2.1 OPTIMIZATION IN THE WASSERSTEIN SPACE

Sampling algorithms can be considered as optimizing some given functionals in the Wasserstein space as Eq. 1. Generally, they only update particles, which causes the evolution of the particles' distribution. Such an evolution finally affects the objective functional.

In particular, given initial distribution $\boldsymbol{x}_0 \sim p_0(\boldsymbol{x})$ and function class $\mathcal{H}$, suppose the update of $\boldsymbol{x}_t$ is

$$\mathrm{d}\boldsymbol{x}_t = \phi_t(\boldsymbol{x}_t), \tag{2}$$

where $\phi_t : \mathbb{R}^d \to \mathbb{R}^d \in \mathcal{H}$. By the continuity equation, we have the differential equation for $p_t(\boldsymbol{x})$,

$$\frac{\mathrm{d}p_t(\boldsymbol{x})}{\mathrm{d}t} = -\nabla \cdot (p_t(\boldsymbol{x})\phi_t(\boldsymbol{x})).$$

For any suitable functional $\mathcal{F}$ w.r.t. $p_t$, its evolution can be presented as

$$\frac{\mathrm{d}}{\mathrm{d}t}\mathcal{F}(p_t) = \int_{\mathbb{R}^d} \frac{\delta\mathcal{F}}{\delta p}(p_t)\partial_t p_t \mathrm{d}\boldsymbol{x} = \int_{\mathbb{R}^d} \nabla\frac{\delta\mathcal{F}}{\delta p}(p_t) \cdot \phi_t(\boldsymbol{x})\mathrm{d}p_t.$$

where $\delta\mathcal{F}/\delta p$ denotes the $L^2(\mathbb{R}^d)$-functional derivative (Villani, 2009; Duncan et al., 2019). Assuming that $\mathcal{F} = H_{p_*}$, we have

**Proposition 2.1.** *The evolution of KL divergence with Eq. 2 is*

$$\frac{\mathrm{d}H_{p_*}(p_t)}{\mathrm{d}t} = -\int p_t(\boldsymbol{x})\phi_t(\boldsymbol{x})^\top \nabla \ln\frac{p_*(\boldsymbol{x})}{p_t(\boldsymbol{x})}d\boldsymbol{x}. \tag{3}$$

Proposition 2.1 is a direct result of the functional derivative for KL divergence, where $\delta\mathcal{F}/\delta p = \ln p + 1 + \ln p^*$ (Chapter 15 of Villani (2009)). By choosing $\phi_t$ which decreases KL divergence via Eq. 3, we can optimize $p_t$ to approach $p_*$.

## 2.2 Log-Sobolev inequality

In the Wasserstein space optimization literature, LSI is particularly crucial to obtaining the convergence rate, which is an analogue to Polyak-Lojasiewicz (PL) inequality in Euclidean space. LSI applies to a wider class of measures than log-concave distributions and can be checked by Bakry-Emery criterion Bakry & Émery (1985). Specifically, bounded perturbation and Lipschitz mapping can preserve the establishment of LSI Vempala & Wibisono (2019), where log-concavity would be failed. For example, subtracting some small Gaussians from a strongly log-concave distribution will destroy the log-concavity of the original distribution, while it still satisfies LSI as long as the Gaussians we subtract are small enough. When the target distribution $p_*$ satisfies $\mu$-LSI, it denotes

$$\mathbb{E}_{p_*}\left[g^2 \ln g^2\right] - \mathbb{E}_{p_*}\left[g^2\right] \ln \mathbb{E}_{p_*}\left[g^2\right] \leq \frac{2}{\mu}\mathbb{E}_{p_*}\left[\|\nabla g\|^2\right], \tag{4}$$

for any differentiable function $g \in L^2(\nu)$. Such an inequality usually provides some connection between the sufficient descent of functional evolution and its exact values.

**Coupling log-Sobolev inequality with Langevin dynamics.** In particular, the most popular algorithm, Langevin dynamics Vempala & Wibisono (2019) chooses

$$\phi_t(\boldsymbol{x}) = \nabla \ln\frac{p_*(\boldsymbol{x})}{p_t(\boldsymbol{x})}, \quad \frac{\mathrm{d}p_t(\boldsymbol{x})}{\mathrm{d}t} = \nabla \cdot \left(p_t(\boldsymbol{x})\nabla \ln\frac{p_t(\boldsymbol{x})}{p_*(\boldsymbol{x})}\right)$$

to decrease the KL functional (Eq. 3) which is equivalent to the particles' update,

$$\mathrm{d}\boldsymbol{x}_t = -\nabla f_*(\boldsymbol{x})\mathrm{d}t + \sqrt{2}\mathrm{d}\boldsymbol{B}_t, \tag{5}$$

where $\boldsymbol{B}_t$ is standard Brownian motion. The introduction of randomness can also convert $\nabla \ln p_t(\boldsymbol{x})$ to a tractable form and the dynamics becomes

$$\frac{\mathrm{d}H_{p_*}(p_t)}{\mathrm{d}t} = -\int p_t(\boldsymbol{x})\left\|\nabla \ln\frac{p_t(\boldsymbol{x})}{p_*(\boldsymbol{x})}\right\|^2 \mathrm{d}\boldsymbol{x} \tag{6}$$

where the absolute value of RHS in Eq. 6 is called relative Fisher information. Taking $g^2 = p_t/p_*$, we have

$$H_{p_*}(p_t) \leq \frac{1}{2\mu}\int p_t(\boldsymbol{x})\left\|\nabla \ln\frac{p_t(\boldsymbol{x})}{p_*(\boldsymbol{x})}\right\|^2 \mathrm{d}\boldsymbol{x}. \tag{7}$$

Then LSI provides a lower bound for gradient norm, leading to sufficient descent for KL divergence. Combining Eq. 7 and Eq. 6, we have

$$\frac{\mathrm{d}H_{p_*}(p_t)}{\mathrm{d}t} \leq -2\mu H_{p_*}(p_t), \tag{8}$$

for some $\mu > 0$. By applying Gronwall's lemma, Eq. 8 yields $H_{p_*}(p_t) \leq H_{p_*}(p_0)\exp(-2\mu t)$, which indicates the linear convergence rate of Langevin dynamics. Nonetheless, coupling LSI with SVGD is challenging due to the introduction of RKHS.

**KL divergence descent with asymptotic SVGD.** As shown in Lemma 3.2 of Liu & Wang (2016), the key point of continuous SVGD is to minimize the RHS of Eq. 3 in the unit ball of the RKHS $\mathcal{H}$ as follows

$$\mathcal{S}(p_t, p_*) = \max_{\phi \in \mathcal{H}} \int p_t(\boldsymbol{x})\phi(\boldsymbol{x})^\top \nabla \ln \frac{p_*(\boldsymbol{x})}{p_t(\boldsymbol{x})} d\boldsymbol{x}, \quad \text{such that } \|\phi\|_{\mathcal{H}} \le 1 \tag{9}$$

where $\mathcal{S}(p_t, p_*)$ is called Kernelized Stein's discrepancy (KSD). Assume that the RKHS is associated with a kernel $k(\boldsymbol{x}, \boldsymbol{y}) : \mathbb{R}^d \times \mathbb{R}^d \to \mathbb{R}$, such that $k(\boldsymbol{x}, \boldsymbol{y}) = \langle \Phi(\boldsymbol{x}), \Phi(\boldsymbol{y}) \rangle_{\mathcal{H}_0}$, and $\mathcal{H}$ is the $d$-times Cartesian product space of $\mathcal{H}_0$ that contains Frobenius-normalized linear functions from $\mathcal{H}$ to $\mathbb{R}^d$, i.e., $\langle \phi_t, \phi_t \rangle_{\mathcal{H}} \le 1$. Note that Eq. 10 defines a functional gradient in RKHS, which indicates the steepest KL decreasing direction with a normalized functional vector. Using integration by parts, Eq. 9 can be rewritten as

$$\phi_t = \arg\max_{\phi \in \mathcal{H}} \int p_t(\boldsymbol{x})(\phi(\boldsymbol{x})^\top \nabla \ln p_*(\boldsymbol{x}) + \nabla\phi(\boldsymbol{x}))d\boldsymbol{x}, \quad \text{such that } \|\phi\|_{\mathcal{H}} \le \mathcal{S}(p_t, q). \tag{10}$$

The explicit form for $\phi_t(\boldsymbol{x})$ is presented as follows.

**Proposition 2.2.** *Assume that $\phi_t$ satisfies Eq. 10. Then we have*

$$\phi_t(\boldsymbol{x}) = \int p_t(\boldsymbol{y}) \left[ \nabla \ln p_*(\boldsymbol{y})k(\boldsymbol{x}, \boldsymbol{y}) + \nabla_1 k(\boldsymbol{y}, \boldsymbol{x}) \right] \mathrm{d}\boldsymbol{y}, \tag{11}$$

*where $\phi_t(\boldsymbol{x})$ can be estimated by particle samples from $p_t$.*

Proposition 2.2 (as Lemma 3.2 proved in Liu & Wang (2016)) makes Eq. 2 become a practically tractable algorithm by Monte Carlo estimation of Eq. 11, where particle samples are from $p_t(\boldsymbol{y})$. Note that Eq. 2 and Eq. 11 naturally lead to the algorithm of SVGD.

Combining Proposition 2.1 and 2.2, the dissipation of the KL divergence along continuous SVGD can be obtained,

$$\frac{\mathrm{d}H_{p_*}(p_t)}{\mathrm{d}t} = - \int_{\mathbb{R}^d} \int_{\mathbb{R}^d} k(\boldsymbol{x}, \boldsymbol{y})p_t(\boldsymbol{x})p_t(\boldsymbol{y}) \cdot \left[ \nabla \ln \frac{p_t(\boldsymbol{x})}{p^*(\boldsymbol{x})} \cdot \nabla \ln \frac{p_t(\boldsymbol{y})}{p^*(\boldsymbol{y})} \right] \mathrm{d}\boldsymbol{y}\mathrm{d}\boldsymbol{x}. \tag{12}$$

When the kernel is strictly positive definite, the RHS of Eq. 12 is negative, leading to the decrease of KL divergence. Unlike the sufficient descent bounded by the functional value in Eq. 7, LSI cannot be conducted on the RHS of Eq. 12 due to the kernelization, which also causes the KL convergence rate to be unknown. Instead, previous works (Liu, 2017; Lu et al., 2019) provided an $\mathcal{O}(1/t)$ KSD convergence as

$$\min_{0 \le s \le t} \mathcal{S}(p_t, p_*) \le \frac{1}{t} \int_0^t \mathcal{S}(p_s, p_*)\mathrm{d}s \le \frac{H_{p_*}(p_0)}{t}. \tag{13}$$

## 3 KL CONVERGENCE OF SVGD

In this section, we first show, compared with KSD convergence proved in most previous works on SVGD Liu (2017); Lu et al. (2019); Korba et al. (2020); Salim et al. (2021; 2022), KL convergence is more powerful in explaining the practical performance of real-world applications. After that, we explain Stein log-Sobolev Inequality (SLSI) the necessary condition for analyzing KL convergence of SVGD can hardly be verified. In order to investigate more reasonable conditions, our assumptions are proposed and validated empirically in some simple cases. Finally, we provide the main theorem, i.e., the KL convergence of SVGD under these mild assumptions. Due to the page limit, we left the comparison with previous works by list in Appendix A.

### 3.1 SAMPLING TASKS REQUIRE KL CONVERGENCE

Sampling algorithms are widely used in real-world applications for solving corresponding machine learning problems. Although different tasks usually require different criteria for convergence analysis, most of these criteria can be deduced by KL convergence.

In Bayesian learning Welling & Teh (2011), people expect to capture parameter uncertainty via Markov chain Monte Carlo (MCMC) techniques. Specifically, they prefer to sample the parameter vector $\boldsymbol{w}$ from a posterior distribution presented as

$$p(\boldsymbol{w}|\boldsymbol{z}) \propto p(\boldsymbol{w}) \prod_{i=1}^{n} p(\boldsymbol{z}_i|\boldsymbol{w}),$$

where $\boldsymbol{z} := \{\boldsymbol{z}_i\}_{i=1}^{N}$ denotes a given dataset. The update of $\boldsymbol{w}$ in (Neal et al., 2011)

$$\Delta\boldsymbol{w} = \left(\eta_t \nabla \log p(\boldsymbol{w}) + \eta_t \sum_{i=1}^{N} \nabla \log p(\boldsymbol{z}_i|\boldsymbol{w}) + \sqrt{2\eta_t} \int_0^1 \mathrm{d}\boldsymbol{B}_t\right), \tag{14}$$

is equivalent to minimizing the KL divergence w.r.t between $p(\boldsymbol{w}|\boldsymbol{z})$ and $p_*(\boldsymbol{w}|\boldsymbol{z})$ where

$$p_*(\boldsymbol{w}|\boldsymbol{z}) = \arg\min_{p'} H_{p(\boldsymbol{w}) \prod_{i=1}^{n} p(\boldsymbol{z}_i|\boldsymbol{w})}(p').$$

It means the convergence of Bayesian learning is directly dependent on KL convergence. Another important application of sampling algorithms is to minimize the expected excess risk as

$$\mathbb{E}\left[F(\hat{\mathbf{w}})\right] - F^* \tag{15}$$

where $F$ denotes the objective function of the stochastic optimization under the unknown data distribution $P$

$$F(w) = \mathbb{E}_P\left[f(w, \mathbf{z})\right] = \int_z f(w, z) P(\mathrm{d}z),$$

and $F^*$ denotes $\inf_{w \in \mathbb{R}^d} F(w)$. When $F$ is $L$-smooth, previous gradient-based MCMC methods would like to analyze the convergence with the general framework by

$$\begin{aligned}
\mathbb{E}\left[F(\hat{\mathbf{w}}_k)\right] - F^* &= \mathbb{E}\left[F(\hat{\mathbf{w}}_k)\right] - \mathbb{E}\left[F(\hat{\mathbf{w}}^*)\right] + \mathbb{E}\left[F(\hat{\mathbf{w}}^*)\right] - F^* \\
&= \int P^{(n)}(\mathrm{d}\boldsymbol{z}) \left[\int_{\mathbb{R}^d} F(\boldsymbol{w})\tilde{p}_{k,\boldsymbol{z}}(\boldsymbol{w})\mathrm{d}\boldsymbol{w} - \int_{\mathbb{R}^d} F(\boldsymbol{w})p_{*,\boldsymbol{z}}(\boldsymbol{w})\mathrm{d}\boldsymbol{w}\right] + \mathbb{E}\left[F(\hat{\mathbf{w}}^*)\right] - F^* \\
&\leq \int P^{(n)}(\mathrm{d}\boldsymbol{z}) \underbrace{\left[LC \cdot W_2(\tilde{p}_{k,\boldsymbol{z}}, p_{*,\boldsymbol{z}})\right]}_{\text{Training error}} + \mathbb{E}\left[F(\hat{\mathbf{w}}^*)\right] - F^*
\end{aligned} \tag{16}$$

where the $n$-tuple (data) $\boldsymbol{z} = \{z_1, z_2, \ldots, z_n\}$ of i.i.d. samples are drawn from $P$. It means the minimization of Wasserstein 2 distance between $\tilde{p}_{t,\boldsymbol{z}}$ (MCMC samples at time $t$) and $p_{*,\boldsymbol{z}}(x) \propto p(\boldsymbol{w}|\boldsymbol{z})$ leads to convergence of expected excess risk. The aforementioned results can be directly deduced by the KL convergence (Raginsky et al., 2017; Xu et al., 2018) with

$$W_2(\tilde{p}_{k,\boldsymbol{z}}, p_{*,\boldsymbol{z}}) \leq C \cdot \left(\sqrt{H_{p_{*,\boldsymbol{z}}}(\tilde{p}_{k,\boldsymbol{z}})} + \left(\frac{H_{p_{*,\boldsymbol{z}}}(\tilde{p}_{k,\boldsymbol{z}})}{2}\right)^{1/4}\right).$$

Unfortunately, the connection between $W_2(\tilde{p}_{k,\boldsymbol{z}}, p_{*,\boldsymbol{z}})$ and $\mathcal{S}(\tilde{p}_{k,\boldsymbol{z}}, p_{*,\boldsymbol{z}})$ depends on the choice of RKHS, which is highly specialized and non-general. From a theoretical perspective, when the RKHS is over-smooth with a sufficient large bandwidth, $k(\boldsymbol{x}, \boldsymbol{y}) = \sigma^{-d} \exp(-\|\boldsymbol{x} - \boldsymbol{y}\|^2/2\sigma^2)$ with large $\sigma$, the corresponding kernelized gradient tend to diminish, i.e., $\lim_{\sigma \to \infty} \mathbb{E}_{p_t}[k(\boldsymbol{x}, \boldsymbol{y})\nabla \ln \frac{p_t(y)}{p_*(y)}] = 0$. That means the KSD can be arbitrarily small with improper RKHS choices. In this condition, the convergence of KSD does not make much sense about the quality of $p_t$, since it can be simply controlled by some special RKHS.

## 3.2 KL CONVERGENCE OF SVGD WITH DIFFERENT ASSUMPTIONS

To investigate KL convergence of SVGD, some previous works (Duncan et al., 2019; Korba et al., 2020; Salim et al., 2021) introduce the following assumption.

**Assumption 1.** *The probability density $p_*$ satisfies Stein log-Sobolev inequality (SLSI) with a constant $\mu > 0$, if for any $p_t \in \mathcal{P}_2(\mathbb{R}^d)$, it has*

$$H_{p_*}(p_t) \leq \frac{1}{2\mu}\mathcal{S}(p_t, p_*). \tag{17}$$

We can immediately obtain

$$H_{p_*}(p_t) \leq \frac{1}{2\mu} \int_{\mathbb{R}^d} \int_{\mathbb{R}^d} k(\boldsymbol{x}, \boldsymbol{y}) p_t(\boldsymbol{x}) p_t(\boldsymbol{y}) \cdot \left[ \nabla \ln \frac{p_t(\boldsymbol{x})}{p_*(\boldsymbol{x})} \cdot \nabla \ln \frac{p_t(\boldsymbol{y})}{p_*(\boldsymbol{y})} \right] \mathrm{d}\boldsymbol{y} \mathrm{d}\boldsymbol{x}, \qquad (18)$$

and a linear KL convergence can be achieved by combining Eq. 12 and Eq. 18 for SVGD (Duncan et al., 2019; Korba et al., 2020; Salim et al., 2021). However, the verification of this assumption is highly non-trivial because we cannot test all $p_t \in \mathcal{P}_2(\mathbb{R}^d)$. Only when the designed RKHS is overly regular, the RHS of Eq. 18 can be estimated. In the meanwhile, an overly regular kernel, i.e.,

$$\int p_*(\boldsymbol{x}) \nabla \ln p_*(\boldsymbol{x}) \cdot \nabla \ln p_*(\boldsymbol{x}) k(\boldsymbol{x}, \boldsymbol{x}) - 2\nabla \ln p_*(\boldsymbol{x}) \cdot \nabla_1 k(\boldsymbol{x}, \boldsymbol{x}) + \nabla_1 k(\boldsymbol{x}, \boldsymbol{x}) \cdot \nabla_2 k(\boldsymbol{x}, \boldsymbol{x}) \mathrm{d}\boldsymbol{x} < \infty,$$
$$(19)$$

will make SLSI fail, which is indicated in (Duncan et al., 2019). Besides, Eq. 19 holds for the most widely used smoothing kernels, such as Radial basis function (RBF) kernel. The contradiction between Eq. 18 and Eq. 19 makes the current analysis in KL divergence highly restricted.

In this condition, we expect more reasonable assumptions to investigate KL convergence of SVGD. Similar to (Arbel et al., 2019), we have additional assumptions on trajectory of $p_t$. Specifically, we assume the following.

[**A₁**] $p_*$ satisfies $\mu$-log-Sobolev Inequality (Eq. 4) and $f_*$ is $L$-smooth, i.e., for any $\boldsymbol{x}, \boldsymbol{y} \in \mathbb{R}^d$, $\|\nabla f_*(\boldsymbol{x}) - \nabla f_*(\boldsymbol{y})\| \leq L\|\boldsymbol{x} - \boldsymbol{y}\|$.

[**A₂**] $f_t$ is $L$-smooth where $p_t = e^{-f_t}$.

[**A₃**] $p_t$ is warm: $\sup_{\boldsymbol{x} \in \mathbb{R}^d} p_t(\boldsymbol{x})/p_*(\boldsymbol{x}) \leq \beta$ for some constant $\beta \geq 1$.

Assumption [**A₁**] and [**A₂**] are similar to the convexity geometry and $L$-smoothness in conventional Euclidean optimization. Assumption [**A₃**] restricts the domain of our proof: the tail of $p_t$ should be lighter than $p_*$, which is widely used in Langevin dynamics. Compared with SLSI, due to the decoupling of requirements of the target distribution $p_*$ and designed kernels $k$, we can verify these assumptions in several ways. The establishment of LSI of the target distribution $p_*$ can be checked by the criterion mentioned in Section 2.2. For the trajectory assumptions, i.e., [**A₂**] and [**A₃**], we provide the empirical validation by showing the estimation of density ratio and smoothness of $p_t$ in some simple cases with the growth of $t$ (Fig 1). Then, we have the following theorem.

**Theorem 3.1.** *Suppose Assumption [A₁]-[A₃] are satisfied, and chi-square $D_\chi(p_0, p_*) \leq 1/4$. For any $\epsilon > 0$, if we set reweighted kernel $k$:*

$$k(\boldsymbol{x}, \boldsymbol{y}) = (p_*(\boldsymbol{x}))^{-1/2} \overline{k}_\sigma(\boldsymbol{x}, \boldsymbol{y}) (p_*(\boldsymbol{y}))^{-1/2}, \quad \overline{k}_\sigma(\boldsymbol{x}, \boldsymbol{y}) = \tilde{k}_\sigma(\boldsymbol{x} - \boldsymbol{y}) = \sigma^{-d} \tilde{k}(\sigma^{-1}(\boldsymbol{x} - \boldsymbol{y})),$$

$$\int_{\mathbb{R}^d} \|\boldsymbol{y}\|^4 \cdot \tilde{k}(\boldsymbol{y}) \mathrm{d}\boldsymbol{y} \leq M, \quad \text{and} \quad \left| 1 - \int_{\mathbb{R}^d} \overline{k}_\sigma(\boldsymbol{x}, \boldsymbol{x} - \boldsymbol{y}) \mathrm{d}\boldsymbol{y} \right| \leq \frac{1}{2\sqrt{2}},$$
$$(20)$$

*where*

$$\sigma = \min \left( 1, \frac{\epsilon}{12LM\sqrt{\beta}} \cdot \left( 16Cd + \frac{9\beta Cd}{2M} + 6\sqrt{\beta}L + 3CL + \beta CL \right)^{-1} \right), \qquad (21)$$

$C = \int \sqrt{p_*(\boldsymbol{x})} \mathrm{d}\boldsymbol{x}$ , *then the KL divergence between $p_t$ and $p_*$ satisfies*

$$H_{p_*}(p_t) \leq \max \left( 0, \left( H_{p_*}(p_0) - \frac{64\epsilon}{\mu} \right) \right) \cdot \exp \left( -\frac{\mu t}{16} \right) + \frac{64\epsilon}{\mu}. \qquad (22)$$

**Remark 1.** *It should be noted that $C < \infty$ is proven in Lemma B.1. Although we require the trajectory of the algorithm to satisfy some assumptions, the actual requirements are much looser. For example, we allow the coefficients $L$ of smoothness in Assumption [A₂] and the maximum density ratio $\beta$ in Assumption [A₃] to increase with the number of iteration $t$ growth. Even if the rate of growth is polynomial, $O(1/\epsilon)$ convergence rate can still be obtained by decreasing $\sigma_t$ in the reweighted kernel in Eq. 21, which is shown in Remark 3. Assumption [A₃] is actually introduced to control Rényi divergence between $p_t$ and $p_*$ will not be infinity in some region near the target, which can be easily obtained in Langevin dynamics.*

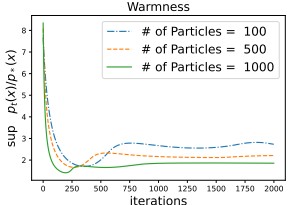 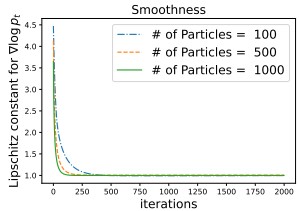

Figure 1: Illustration of warmness and smoothness evolution, where $p_0 \sim \mathcal{N}(0, 0.25)$, $p_* \sim \mathcal{N}(0, 5)$.

This theorem demonstrates that, by introducing reweighted kernel $k(\boldsymbol{x}, \boldsymbol{y})$ and controlling the variance $\sigma$ of smoothing kernel $\tilde{k}$, SVGD initialized in a local region will provide a linear convergence rate to any $\epsilon$-neighborhood of the target distribution. To validate this result, we provide experiments in synthetic data in Appendix D, and illustrate SVGD with reweighted kernels usually achieves a lower KL divergence compared with traditional SVGD. It should be noticed that commonly used kernels, e.g., RBF kernel and Bump kernel, are proper $\tilde{k}$. Besides, the linear convergence shows all the parameters will not deteriorate the convergence of SVGD when $\sigma$ is small enough.

## 4 REWEIGHTED KERNEL FOR KL CONVERGENCE

In this section, we mainly explain why we should introduce reweighted kernels in Theorem 3.1. The intuition can be split into 2 parts: (1) the infeasibility of the usage of LSI due to the kernel approximation error; (2) the tractable kernel approximation error form with a reweighted kernel.

### 4.1 KERNEL APPROXIMATION

Intuitively, to measure the error between Wasserstein gradient and its kernelized one, the most direct idea is to control the error of relative Fisher information and make use of LSI. However, this idea will encounter some fatal bottlenecks.

**The bottleneck of SVGD analysis with Eq. 7** If we directly upper bound the descent of KL divergence by Eq. 7, we have

$$\frac{\mathrm{d}H_{p_*}(p_t)}{\mathrm{d}t} \leq \underbrace{-\frac{1}{2} \int p_t(\boldsymbol{x}) \left\| \nabla \ln \frac{p_t(\boldsymbol{x})}{p_*(\boldsymbol{x})} \right\|^2 \mathrm{d}\boldsymbol{x}}_{\text{sufficient descent}}$$

$$+ \underbrace{\frac{1}{2} \int_{\mathbb{R}^d} p_t(\boldsymbol{x}) \left\| \int p_t(\boldsymbol{y}) k(\boldsymbol{x}, \boldsymbol{y}) \nabla \ln \frac{p_*(\boldsymbol{y})}{p_t(\boldsymbol{y})} \mathrm{d}\boldsymbol{y} - \nabla \ln \frac{p_*(\boldsymbol{x})}{p_t(\boldsymbol{x})} \right\|^2 \mathrm{d}\boldsymbol{x}}_{\text{kernel approximation error}}, \tag{23}$$

where the kernel approximation error can hardly be upper bounded due to $p_t(\boldsymbol{y})$ in integration. It means if we directly plug some smoothing kernels into the iteration of SVGD, kernel approximate error may dominate RHS of Eq. 23 and cause SVGD to converge to a limit different from the target distribution with an uncontrollable bias.

**Failures of Smoothing Kernels.** Smoothing Kernels (kernel smoothers), such as radial basis function kernel, are widely used in SVGD, due to their universal approximation capability to smooth functions (Park & Sandberg, 1991; Micchelli et al., 2006). Assume that $\bar{k}_\sigma(\boldsymbol{x}, \boldsymbol{y}) = \tilde{k}_\sigma(\boldsymbol{x} - \boldsymbol{y}) = \sigma^{-d} \tilde{k}(\sigma^{-1}(\boldsymbol{x} - \boldsymbol{y}))$ is a smoothing kernel with parameter $\sigma > 0$, where $\sigma$ is called the bandwidth of the kernel. The variance $\sigma^2$ in the smoothing kernel tends to control the smoothness of the estimated gradient. A large $\sigma$ makes kernelized gradient well-estimated with finite samples while the kernel approximation error is large. In the population limit, where randomness from $p_t(\boldsymbol{y})$ is ignored, the optimal smoothing kernel should be Dirac delta function $\delta_{\boldsymbol{x}}(\boldsymbol{y})$, where the kernel bandwidth is sufficiently small to estimate Wasserstein gradient. For each point $\boldsymbol{x}$, the kernelized gradient becomes $p_t(\boldsymbol{x}) \nabla \ln \frac{p_*(\boldsymbol{x})}{p_t(\boldsymbol{x})}$, which means that for those low-density areas $p_t(\boldsymbol{x}) \to 0$, the kernelization suffers

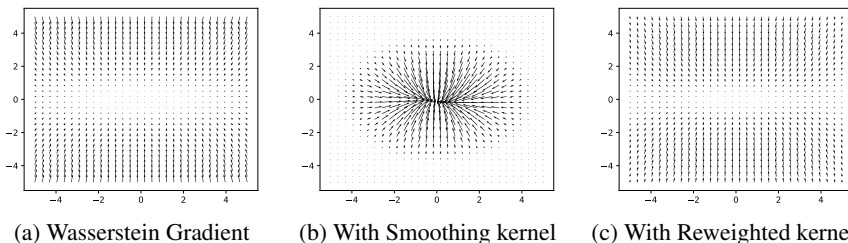

(a) Wasserstein Gradient      (b) With Smoothing kernel      (c) With Reweighted kernel

Figure 2: Illustration of gradient vanishing with smoothing kernel. The vector field illustrates $-\nabla \ln p_t(\boldsymbol{x})$ for data distribution $\mathcal{N}(0, \operatorname{diag}(1, 0.5^2))$

from gradient vanishing. Such biased Wasserstein gradient estimation in smoothing kernel SVGD will have an additional $p_t$ term, which hampers the sufficient descent of each iteration and even the convergence rate. This indicates that SVGD is not compatible with smoothing kernels in low-density parts. Ideally, the kernel should be reweighted by some $1/\sqrt{p_t(\boldsymbol{x})p_t(\boldsymbol{y})}$, which can balance the vanishing scaling in the current SVGD. However, $p_t$ is unknown in general, so this reweighting cannot provide algorithmic insight to improve SVGD.

To solve this problem, a very intuitive idea is to balance the order of $p_t$, we may require kernel $k$ to be related to $p_t$ through the following proposition.

**Proposition 4.1.** *If we use the kernel $k(\boldsymbol{x}, \boldsymbol{y}) = (p_t(\boldsymbol{x}))^{-1/2} \, \overline{k}_\sigma(\boldsymbol{x}, \boldsymbol{y}) \, (p_t(\boldsymbol{y}))^{-1/2}$, by choosing delta function as the smoothing kernel $\overline{k}_0(\boldsymbol{x}, \boldsymbol{y}) = \delta_{\boldsymbol{x}}(\boldsymbol{y})$, kernel approximation error is $0$ and SVGD is equivalent to the Wasserstein gradient flow.*

Unfortunately, the reweighting strategy with $p_t(\boldsymbol{x})$ makes the iteration of SVGD (Eq. 11) computationally intractable as $p_t(\boldsymbol{x})$ and $\nabla p_t(\boldsymbol{x})$ are unknown in general. If we consider the local convergence of SVGD, when $p_t(\boldsymbol{x})$ is approaching $p_*(\boldsymbol{x})$, we can expect a lower kernel approximation error by replacing $p_t^{-1/2}$ in Proposition 4.1 with $p_*^{-1/2}(\cdot)$ as follows

$$k(\boldsymbol{x}, \boldsymbol{y}) = (p_*(\boldsymbol{x}))^{-1/2} \, \overline{k}_\sigma(\boldsymbol{x}, \boldsymbol{y}) \, (p_*(\boldsymbol{y}))^{-1/2} \tag{24}$$

where $\overline{k}_\sigma$ denotes the smoothing kernel, and it satisfies

$$\int_{\mathbb{R}^d} \|\boldsymbol{y}\|^4 \cdot \overline{k}_\sigma(\boldsymbol{x}, \boldsymbol{x} - \boldsymbol{y}) \mathrm{d}\boldsymbol{y} < \infty \quad \text{and} \quad \left| 1 - \int_{\mathbb{R}^d} \overline{k}_\sigma(\boldsymbol{x}, \boldsymbol{x} - \boldsymbol{y}) \mathrm{d}\boldsymbol{y} \right| \leq \frac{1}{2\sqrt{2}}. \tag{25}$$

Notice that many popular kernels satisfies Eq. 25, e.g., standard RBF kernel, Bump function (Eq. 26), etc.

$$\tilde{k}_\sigma(\boldsymbol{z}) = \frac{1}{\sigma} \exp\left( -\frac{1}{1 - \|\boldsymbol{z}\|^2/\sigma^2} \right) \quad \|\boldsymbol{z}\| \in \mathcal{B}(\boldsymbol{0}, \sigma). \tag{26}$$

Therefore, the dynamics of KL divergence in such a reweighted kernel is

$$\frac{\mathrm{d}H_{p_*}(p_t)}{\mathrm{d}t} = -\int_{\mathbb{R}^d} \int_{\mathbb{R}^d} \overline{k}_\sigma(\boldsymbol{x}, \boldsymbol{y}) \cdot \left[ \frac{p_t(\boldsymbol{x})}{\sqrt{p_*(\boldsymbol{x})}} \nabla \ln \frac{p_t(\boldsymbol{x})}{p_*(\boldsymbol{x})} \cdot \frac{p_t(\boldsymbol{y})}{\sqrt{p_*(\boldsymbol{y})}} \nabla \ln \frac{p_t(\boldsymbol{y})}{p_*(\boldsymbol{y})} \right] \mathrm{d}\boldsymbol{y}\mathrm{d}\boldsymbol{x}. \tag{27}$$

Similar to the requirement of a delta function in 4.1, a small $\sigma$ is preferred in our setting, since our analysis is based on the population limit of $p_t(\boldsymbol{x})$. Besides, our analysis would indicate the kernel choice to obtain the convergence.

Compared with utilizing smoothing kernel directly, we may expect a smaller kernel approximation error by introducing kernels as Eq. 24. We also validate this phenomenon by Figure 2, which has shown the gradient vanishing phenomenon with the smoothing kernel. Figure 2 (a) is the vector field of Wasserstein gradient $\nabla \ln p_*(x)/p_t(x)$, which is a linear function in Gaussian case. However, when using a smoothing kernel the low-density area in Figure 2 (b) has almost no gradient, which makes the particle in this area stuck. Our proposed reweighted kernel amplifies the gradient in low-density areas, the resulting gradient is similar to the Wasserstein gradient.

## 4.2 ERROR REWEIGHTING

To achieve a better kernel approximation error with reweighted smoothing kernel Eq. 24, we require a corresponding local version of log-Sobolev inequality to control the sufficient descent of the evolution of the KL divergence (Eq. 27).

**Local version of log-Sobolev inequality.** In this context, the sufficient descent term should be reformulated to obtain a well-behaved kernel approximation error. The choice of $g$ in Eq. 38 may convert the kernel approximation error to a tractable form. By choosing $g(\boldsymbol{x}) = \frac{p_t(\boldsymbol{x})}{p_*(\boldsymbol{x})}$, we can find the upper bound of KL divergence by $\chi^2$ version of Rényi information.

**Lemma 4.2.** *Suppose $p_*$ satisfies the $\mu$-log-Sobolev inequality (LSI) with a constant $\mu > 0$. When any probability density function $p_t$ satisfies $D_{\chi^2}(p_t, p_*) \leq 1/2$, we have*

$$\frac{\mu}{4} H_{p_*}(p_t) \leq \int_{\mathbb{R}^d} p_*(\boldsymbol{x}) \cdot \left\| \frac{p_t(\boldsymbol{x})}{p_*(\boldsymbol{x})} \nabla \ln \frac{p_t(\boldsymbol{x})}{p_*(\boldsymbol{x})} \right\|^2 \mathrm{d}\boldsymbol{x}. \tag{28}$$

Lemma 4.2 indicates that the KL divergence is also bounded by the Wasserstein gradient of $\chi^2$ divergence. If $p_t/p_*$ is bounded, Eq. 28 provides a tighter upper bound of KL divergence compared with that in Eq. 7, especially for the tail part. Thus, controlling the error term in this form has more potential. With such a construction, the decreasing of KL divergence satisfies

$$\frac{\mathrm{d}}{\mathrm{d}t} H_{p_*}(p_t) \leq \underbrace{-\frac{1}{2} \int_{\mathbb{R}^d} p_*(\boldsymbol{x}) \cdot \left\| \frac{p_t(\boldsymbol{x})}{p_*(\boldsymbol{x})} \nabla \ln \frac{p_t(\boldsymbol{x})}{p_*(\boldsymbol{x})} \right\|^2 \mathrm{d}\boldsymbol{x}}_{\text{sufficient descent}}$$
$$+ \frac{1}{2} \underbrace{\int_{\mathbb{R}^d} \left\| \frac{p_t(\boldsymbol{x})}{\sqrt{p_*(\boldsymbol{x})}} \nabla \ln \frac{p_t(\boldsymbol{x})}{p_*(\boldsymbol{x})} - \int_{\mathbb{R}^d} \overline{k}_\sigma(\boldsymbol{x}, \boldsymbol{y}) \frac{p_t(\boldsymbol{y})}{\sqrt{p_*(\boldsymbol{y})}} \nabla \ln \frac{p_t(\boldsymbol{y})}{p_*(\boldsymbol{y})} \mathrm{d}\boldsymbol{y} \right\|^2 \mathrm{d}\boldsymbol{x}}_{\text{kernel approximation error}}. \tag{29}$$

**Lemma 4.3.** *Assume that $\ln p_*$ and $\ln p_t$ are $L$-smooth; $p_t$ is warm: $\sup_{\boldsymbol{x} \in \mathbb{R}^d} p_t(\boldsymbol{x})/p_*(\boldsymbol{x}) \leq \beta$ and $H_{p_*}(p_0) \leq (2\beta)^{-1}$. Then by choosing $\sigma = \min\left(1, \mathcal{O}\left(\frac{\epsilon}{\beta^{1.5}L + \beta L^2}\right)\right)$, we have*

$$\int_{\mathbb{R}^d} \left\| \frac{p_t(\boldsymbol{x})}{\sqrt{p_*(\boldsymbol{x})}} \nabla \ln \frac{p_t(\boldsymbol{x})}{p_*(\boldsymbol{x})} - \int_{\mathbb{R}^d} \overline{k}_\sigma(\boldsymbol{x}, \boldsymbol{y}) \frac{p_t(\boldsymbol{y})}{\sqrt{p_*(\boldsymbol{y})}} \nabla \ln \frac{p_t(\boldsymbol{y})}{p_*(\boldsymbol{y})} \mathrm{d}\boldsymbol{y} \right\|^2 \mathrm{d}\boldsymbol{x} \leq 4\epsilon + \frac{1}{4} \int_{\mathbb{R}^d} p_*(\boldsymbol{x}) \cdot \left\| \frac{p_t(\boldsymbol{x})}{p_*(\boldsymbol{x})} \nabla \ln \frac{p_t(\boldsymbol{x})}{p_*(\boldsymbol{x})} \right\|^2 \mathrm{d}\boldsymbol{x},$$

*where $\overline{k}_\sigma$ satisfies requirements in Eq. 25.*

In this condition, we nearly establish the "Stein log-Sobolev Inequality" with arbitrary small $\epsilon$ by combining Eq. 27, Eq. 28, and Eq. 29 as follows

$$\frac{\mu}{8} H_{p_*}(p_t) \leq \int_{\mathbb{R}^d} \int_{\mathbb{R}^d} \overline{k}_\sigma(\boldsymbol{x}, \boldsymbol{y}) \cdot \left[ \frac{p_t(\boldsymbol{x})}{\sqrt{p_*(\boldsymbol{x})}} \nabla \ln \frac{p_t(\boldsymbol{x})}{p_*(\boldsymbol{x})} \cdot \frac{p_t(\boldsymbol{y})}{\sqrt{p_*(\boldsymbol{y})}} \nabla \ln \frac{p_t(\boldsymbol{y})}{p_*(\boldsymbol{y})} \right] \mathrm{d}\boldsymbol{y}\mathrm{d}\boldsymbol{x} + 4\epsilon. \tag{30}$$

Therefore, reweighted kernels can be considered as a sufficient condition for establishing local "Stein log-Sobolev Inequality" near the target distribution $p_*$.

## 5 CONCLUSIONS

In this paper, we prove the local linear convergence for SVGD with reweighted kernel in KL divergence. In particular, our analysis is based on the smoothing kernel for SVGD algorithm and we point out that the conventional smoothing kernel fails to provide valid gradient scaling in low-density areas. Thus, we highlight that the reweighting is necessary for smoothing kernels in SVGD algorithm. With $(p_*(\boldsymbol{x})p_*(\boldsymbol{y}))^{-1/2}$ weighting for $k(\boldsymbol{x}, \boldsymbol{y})$, we provides the KL convergence rate for SVGD algorithm locally for log-Sobolev $p_*(\boldsymbol{x})$. Our analysis provide new insights on the kernel design in SVGD, especially for gradient amplification in the low-density area.

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

## A CONVERGENCE RATE COMPARISON IN SECTION 3

The assumptions are listed as follows

**[AS$_1$]** $p_*$ satisfies $\mu$-log-Sobolev Inequality (Eq. 4).

**[AS$_2$]** $p_*$ satisfies Talagrand 1 Inequality.

**[AS$_3$]** The Stein log-Sobolev inequality with constant $\lambda$, i.e., $\mathrm{KL}\,(p\|p^*) \leq \frac{1}{2\lambda}\mathbb{D}\,(p\|p^*)^2$

**[AS$_4$]** $f_*$ is $L$-smooth, i.e., for any $\boldsymbol{x}, \boldsymbol{y} \in \mathbb{R}^d$,
$$\|\nabla f_*(\boldsymbol{x}) - \nabla f_*(\boldsymbol{y})\| \leq L\|\boldsymbol{x} - \boldsymbol{y}\|.$$

**[AS$_5$]** $f_t$ is $L$-smooth where $p_t = e^{-f_t}$.

**[AS$_6$]** $p_t$ is warm: $\sup_{\boldsymbol{x} \in \mathbb{R}^d} p_t(\boldsymbol{x})/p_*(\boldsymbol{x}) \leq \beta$ for some constant $\beta \geq 1$.

**[AS$_7$]** $p_t$ SVGD satisfy:
$$\int k(\boldsymbol{x}, \boldsymbol{x}) p_t(\boldsymbol{x}) d\boldsymbol{x} < \infty. \tag{31}$$

**[AS$_8$]** Kernel regularization assumption 1:
$$\sup_{\boldsymbol{x}} \left\{ \frac{1}{2} \|\nabla \log p^*\|_{\mathrm{Lip}}\, k(\boldsymbol{x}, \boldsymbol{x}) + 2\nabla_{\boldsymbol{x}\boldsymbol{x}'} \right\} < \infty. \tag{32}$$

**[AS$_9$]** Kernel regularization assumption 2:
$$\|k(\boldsymbol{x}, \cdot)\|_{\mathcal{H}} \leq B \quad \text{and} \quad \|\nabla_{\boldsymbol{x}} k(\boldsymbol{x}, \cdot)\|_{\mathcal{H}^d} \leq B \tag{33}$$

**[AS$_1$0]** Bounded moment assumptions:
$$\sup_t \int \|\boldsymbol{x}\|\, p_t(\boldsymbol{x})\mathrm{d}\boldsymbol{x} < \infty. \tag{34}$$

We compare our theoretical results with all previous work where Stein discrepancy is abbreviated as SD.

Table 1: Comparison of convergence rate for sampling algorithms

| Algorithm | Assumptions | Criterion | Asymptotic | Rate |
|---|---|---|---|---|
| ULA Vempala & Wibisono (2019) | **[AS$_1$],[AS$_4$]** | KL | No | $\mathcal{O}(e^{-t})$ |
| SVGD Liu (2017) | **[AS$_7$]** | KL | Yes | N/A |
| SVGD Liu (2017) | **[AS$_4$],[AS$_7$],[AS$_8$]** | SD | Yes | $\mathcal{O}(1/t)$ |
| SVGD Korba et al. (2020) | **[AS$_3$],[AS$_4$],[AS$_7$]** | KL | Yes | $\mathcal{O}(e^{-t})$ |
| SVGD Korba et al. (2020) | **[AS$_4$],[AS$_7$],[AS$_9$],[AS$_1$0]** | SD | No | $\mathcal{O}(1/t)$ |
| SVGD Salim et al. (2021) | **[AS$_4$],[AS$_2$],[AS$_7$],[AS$_9$]** | SD | No | $\mathcal{O}(1/t)$ |
| Ours | **[AS$_1$],[AS$_4$],[AS$_5$],[AS$_6$]** | KL | Yes | $\mathcal{O}(e^{-t})$ |

## B IMPORTANT LEMMAS IN SECTION 4

In order to facilitate the lemma in Chapter 4, we first revisit the dynamics of SVGD.

**Remark 2.** *With a slight abuse of notation, $p_t$ of SVGD follows the continuity equation:*
$$\partial_t p_t + \nabla \cdot (p_t \phi_t) = 0 \tag{35}$$
*in the sense of distribution (rather than almost sure), where*
$$\phi_t(\boldsymbol{x}) = \int (k(\boldsymbol{x}, \boldsymbol{y}) \nabla \ln p_*(\boldsymbol{y}) + \nabla_{\boldsymbol{y}} k(\boldsymbol{x}, \boldsymbol{y})) \, \mathrm{d}p_t(\boldsymbol{y}). \tag{36}$$
*Notice that Eq. 35 holds* in the sense of distribution *here means*
$$\int \partial_t p_t(\boldsymbol{x}) v(\boldsymbol{x})\mathrm{d}\boldsymbol{x} = \int \nabla v(\boldsymbol{x}) \cdot \phi_t(\boldsymbol{x})\mathrm{d}p_t(\boldsymbol{x}) \tag{37}$$
*for all $v(\boldsymbol{x}) \in C_c^\infty(\mathbb{R}^d)$.*

## B.1 PROOF OF PROPOSITION 4.1

*Proof.* When $k(\boldsymbol{x}, \boldsymbol{y}) = (p_t(\boldsymbol{x}))^{-1/2} \, \overline{k}_\sigma(\boldsymbol{x}, \boldsymbol{y}) \, (p_t(\boldsymbol{y}))^{-1/2}$ and $\overline{k}_0(\boldsymbol{x}, \boldsymbol{y}) = \delta_{\boldsymbol{x}}(\boldsymbol{y})$, we have

$$k(\boldsymbol{x}, \boldsymbol{y}) = \frac{\delta_{\boldsymbol{x}}(\boldsymbol{y})}{(p_t(\boldsymbol{x}))^{1/2} \, (p_t(\boldsymbol{y}))^{1/2}} = \frac{\delta_{\boldsymbol{x}}(\boldsymbol{y})}{p_t(\boldsymbol{y})}.$$

Thus,

$$\phi_t(\boldsymbol{x}) = \int_{\mathbb{R}^d} p_t(\boldsymbol{y}) \frac{\delta_{\boldsymbol{x}}(\boldsymbol{y})}{p_t(\boldsymbol{y})} \nabla \ln \frac{p_t(\boldsymbol{y})}{p_*(\boldsymbol{y})} \mathrm{d}\boldsymbol{y} = \nabla \ln \frac{p_t(\boldsymbol{x})}{p_*(\boldsymbol{x})},$$

which indicates that $\phi_t(\boldsymbol{x})$ is exactly the Wasserstein gradient and the kernel approximation error is $0$. $\qquad\square$

## B.2 PROOF OF LEMMA 4.2

*Proof.* With LSI, all smooth function $g \colon \mathbb{R}^d \to \mathbb{R}$ with $\mathbb{E}_{p_*}[g^2] < \infty$,

$$\mathbb{E}_{p_*}\left[g^2 \ln g^2\right] - \mathbb{E}_{p_*}\left[g^2\right] \ln \mathbb{E}_{p_*}\left[g^2\right] \le \frac{2}{\mu} \mathbb{E}_{p_*}\left[\|\nabla g\|^2\right]. \tag{38}$$

Suppose $g = p_t/p_*$, we have $\mathbb{E}_{p_*}[g^2] = D_{\chi^2}(p_t, p_*) + 1 < \infty$ and

$$
\begin{aligned}
&\int_{\mathbb{R}^d} p_*(\boldsymbol{x}) \cdot \left(\frac{p_t(\boldsymbol{x})}{p_*(\boldsymbol{x})}\right)^2 \ln \left(\frac{p_t(\boldsymbol{x})}{p_*(\boldsymbol{x})}\right)^2 \mathrm{d}\boldsymbol{x} \\
&\quad - \int_{\mathbb{R}^d} p_*(\boldsymbol{x}) \cdot \left(\frac{p_t(\boldsymbol{x})}{p_*(\boldsymbol{x})}\right)^2 \mathrm{d}\boldsymbol{x} \cdot \ln \left[\int_{\mathbb{R}^d} p_*(\boldsymbol{x}) \cdot \left(\frac{p_t(\boldsymbol{x})}{p_*(\boldsymbol{x})}\right)^2 \mathrm{d}\boldsymbol{x}\right] \\
&\le \frac{2}{\mu} \int_{\mathbb{R}^d} p_*(\boldsymbol{x}) \cdot \left\|\nabla \frac{p_t(\boldsymbol{x})}{p_*(\boldsymbol{x})}\right\|^2 \mathrm{d}\boldsymbol{x} = \frac{2}{\mu} \int_{\mathbb{R}^d} p_*(\boldsymbol{x}) \cdot \left\|\frac{p_t(\boldsymbol{x})}{p_*(\boldsymbol{x})} \nabla \ln \frac{p_t(\boldsymbol{x})}{p_*(\boldsymbol{x})}\right\|^2 \mathrm{d}\boldsymbol{x}.
\end{aligned}
\tag{39}
$$

With the fact $\ln x \le x - 1$ and $\ln x \ge 1 - \frac{1}{x}$ when $x \ge 0$, we have

$$
\begin{aligned}
\ln \left[\int_{\mathbb{R}^d} p_*(\boldsymbol{x}) \cdot \left(\frac{p_t(\boldsymbol{x})}{p_*(\boldsymbol{x})}\right)^2 \mathrm{d}\boldsymbol{x}\right] &\le \int_{\mathbb{R}^d} p_t(\boldsymbol{x}) \cdot \left(\frac{p_t(\boldsymbol{x})}{p_*(\boldsymbol{x})} - 1\right) \mathrm{d}\boldsymbol{x} \\
&= \int_{\mathbb{R}^d} \frac{p_t^2(\boldsymbol{x})}{p_*(\boldsymbol{x})} \cdot \frac{\frac{p_t(\boldsymbol{x})}{p_*(\boldsymbol{x})} - 1}{\frac{p_t(\boldsymbol{x})}{p_*(\boldsymbol{x})}} \mathrm{d}\boldsymbol{x} \le \int_{\mathbb{R}^d} \frac{p_t^2(\boldsymbol{x})}{p_*(\boldsymbol{x})} \cdot \ln \left(\frac{p_t(\boldsymbol{x})}{p_*(\boldsymbol{x})}\right) \mathrm{d}\boldsymbol{x}.
\end{aligned}
\tag{40}
$$

Plugging the previous inequality into LHS of Eq. 39, we have

$$
\begin{aligned}
\mathrm{LHS} &\ge \int_{\mathbb{R}^d} p_*(\boldsymbol{x}) \cdot \left(\frac{p_t(\boldsymbol{x})}{p_*(\boldsymbol{x})}\right)^2 \ln \left(\frac{p_t(\boldsymbol{x})}{p_*(\boldsymbol{x})}\right)^2 \mathrm{d}\boldsymbol{x} \\
&\quad - \int_{\mathbb{R}^d} p_*(\boldsymbol{x}) \cdot \left(\frac{p_t(\boldsymbol{x})}{p_*(\boldsymbol{x})}\right)^2 \mathrm{d}\boldsymbol{x} \cdot \int_{\mathbb{R}^d} \frac{p_t^2(\boldsymbol{x})}{p_*(\boldsymbol{x})} \cdot \ln \left(\frac{p_t(\boldsymbol{x})}{p_*(\boldsymbol{x})}\right) \mathrm{d}\boldsymbol{x} \\
&= 2 \int_{\mathbb{R}^d} p_*(\boldsymbol{x}) \cdot \left(\frac{p_t(\boldsymbol{x})}{p_*(\boldsymbol{x})}\right)^2 \ln \left(\frac{p_t(\boldsymbol{x})}{p_*(\boldsymbol{x})}\right) \mathrm{d}\boldsymbol{x} - \int_{\mathbb{R}^d} p_*(\boldsymbol{x}) \cdot \left(\frac{p_t(\boldsymbol{x})}{p_*(\boldsymbol{x})}\right)^2 \ln \left(\frac{p_t(\boldsymbol{x})}{p_*(\boldsymbol{x})}\right) \mathrm{d}\boldsymbol{x} \\
&\quad - \int_{\mathbb{R}^d} p_*(\boldsymbol{x}) \cdot \left(\left(\frac{p_t(\boldsymbol{x})}{p_*(\boldsymbol{x})}\right)^2 - 1\right) \mathrm{d}\boldsymbol{x} \cdot \int_{\mathbb{R}^d} \frac{p_t^2(\boldsymbol{x})}{p_*(\boldsymbol{x})} \cdot \ln \left(\frac{p_t(\boldsymbol{x})}{p_*(\boldsymbol{x})}\right) \mathrm{d}\boldsymbol{x} \\
&\ge \left(2 - D_{\chi^2}(p_t, p_*) - 1\right) \cdot \int_{\mathbb{R}^d} p_*(\boldsymbol{x}) \cdot \left(\frac{p_t(\boldsymbol{x})}{p_*(\boldsymbol{x})}\right)^2 \ln \left(\frac{p_t(\boldsymbol{x})}{p_*(\boldsymbol{x})}\right) \mathrm{d}\boldsymbol{x} \\
&\ge \frac{1}{2} \int_{\mathbb{R}^d} p_*(\boldsymbol{x}) \cdot \left(\frac{p_t(\boldsymbol{x})}{p_*(\boldsymbol{x})}\right)^2 \ln \left(\frac{p_t(\boldsymbol{x})}{p_*(\boldsymbol{x})}\right) \mathrm{d}\boldsymbol{x},
\end{aligned}
\tag{41}
$$

where the last inequality follows from $D_{\chi^2}(p_t, p_*) \leq 1/2$. Besides, we have

$$
\begin{aligned}
\int_{\mathbb{R}^d} p_*(\boldsymbol{x}) \cdot \left(\frac{p_t(\boldsymbol{x})}{p_*(\boldsymbol{x})}\right)^2 \ln\left(\frac{p_t(\boldsymbol{x})}{p_*(\boldsymbol{x})}\right) \mathrm{d}\boldsymbol{x} &= \int_{\mathbb{R}^d} \frac{p_t(\boldsymbol{x})}{p_*(\boldsymbol{x})} \cdot p_t(\boldsymbol{x}) \ln \frac{p_t(\boldsymbol{x})}{p_*(\boldsymbol{x})} \mathrm{d}\boldsymbol{x} \\
&= \int_{\mathbb{R}^d} \left(\frac{p_t(\boldsymbol{x})}{p_*(\boldsymbol{x})} - 1\right) \cdot p_t(\boldsymbol{x}) \ln \frac{p_t(\boldsymbol{x})}{p_*(\boldsymbol{x})} \mathrm{d}\boldsymbol{x} + \int_{\mathbb{R}^d} p_t(\boldsymbol{x}) \ln \frac{p_t(\boldsymbol{x})}{p_*(\boldsymbol{x})} \mathrm{d}\boldsymbol{x} \geq H_{p_*}(p_t),
\end{aligned}
\tag{42}
$$

where the last inequality follows from $(x - 1) \ln x \geq 0$ for all $x \geq 0$. Combining Eq. 39, Eq. 41 and Eq. 42, we complete the proof, and obtain

$$
\frac{\mu}{4} H_{p_*}(p_t) \leq \int_{\mathbb{R}^d} p_*(\boldsymbol{x}) \cdot \left\| \frac{p_t(\boldsymbol{x})}{p_*(\boldsymbol{x})} \nabla \ln \frac{p_t(\boldsymbol{x})}{p_*(\boldsymbol{x})} \right\|^2 \mathrm{d}\boldsymbol{x}.
\tag{43}
$$

$\square$

## B.3 PROOF OF LEMMA 4.3

Before providing the Kernel Approximation Error (KAE) in Lemma 4.3, we need to introduce some lemmas.

**Lemma B.1.** *Assume that $p(\boldsymbol{x})$ is $L$-smooth and log-Sobolev, then for any $\alpha > 0$*

$$
\int p(\boldsymbol{x})^\alpha \mathrm{d}\boldsymbol{x} < \infty
$$

*Proof.* By Ledoux (1999), when $p(\boldsymbol{x})$ is log-Sobolev, there exists $c > 0$ such that

$$
\int p(\boldsymbol{x}) e^{c\|\boldsymbol{x}\|^2} \mathrm{d}\boldsymbol{x} < \infty
$$

then

$$
\int e^{\ln p(\boldsymbol{x}) + c\|\boldsymbol{x}\|^2} \mathrm{d}\boldsymbol{x} < \infty
$$

Thus, there exists $c_x > 0$, for sufficient large $\|\boldsymbol{x}\| > c_x$,

$$
\ln p(\boldsymbol{x}) < -c\|\boldsymbol{x}\|^2 \quad \text{and} \quad \int_{\|\boldsymbol{x}\| > c_x} p^\alpha(\boldsymbol{x}) \mathrm{d}\boldsymbol{x} < \int_{\|\boldsymbol{x}\| > c_x} e^{-\alpha c\|\boldsymbol{x}\|^2} \mathrm{d}\boldsymbol{x} < \sqrt{\frac{2\pi d}{c\alpha}}
$$

For

$$
\int_{\|\boldsymbol{x}\| \leq c_x} p^\alpha(\boldsymbol{x}) \mathrm{d}\boldsymbol{x} \leq \int_{\|\boldsymbol{x}\| \leq c_x} p^\alpha(\boldsymbol{0}) e^{\alpha L\|\boldsymbol{x}\|^2} \mathrm{d}\boldsymbol{x} < p^\alpha(\boldsymbol{0}) C(d, c_x, \alpha L)
$$

where $C(d, c_x, \alpha L)$ is a constant depend on $d, c_x, \alpha L$. $\square$

**Lemma B.2.** *(A Variant of Lemma. 11 in Vempala & Wibisono (2019)) Suppose $p(\boldsymbol{x}) = e^{-f(\boldsymbol{x})}$, $\boldsymbol{x} \in \mathbb{R}^d$, $f$ is $L$-smooth and $p(\boldsymbol{x})$ satisfies*

$$
\int_{\mathbb{R}^d} \sqrt{p(\boldsymbol{x})} \mathrm{d}\boldsymbol{x} \leq C/2,
$$

*then we have*

$$
\int \sqrt{p(\boldsymbol{x})} \|\nabla f(\boldsymbol{x})\|^2 \mathrm{d}\boldsymbol{x} \leq LCd;
$$

*Proof.* Since $f(\boldsymbol{x})$ is $L$-smooth, we have for any $x$

$$
\nabla^2 f(\boldsymbol{x}) \preceq LI
$$

Using integration by parts, we have

$$\int e^{-f(\boldsymbol{x})/2}\|\nabla f(\boldsymbol{x})\|^2 d\boldsymbol{x} = 4\int e^{-f(\boldsymbol{x})/2}\|\nabla f(\boldsymbol{x})/2\|^2 d\boldsymbol{x} \tag{44}$$

$$= 4\int e^{-f(\boldsymbol{x})/2}\Delta(f(\boldsymbol{x})/2)d\boldsymbol{x} \tag{45}$$

$$\leq 2Ld\int e^{-f(\boldsymbol{x})/2}d\boldsymbol{x} = LCd \tag{46}$$

$\square$

*Proof.* In the following, we mainly focus on providing the upper bound of Kernel Approximation Error (KAE), and have

$$\text{KAE} = \int_{\mathbb{R}^d}\left\|\frac{p_t(\boldsymbol{x})}{\sqrt{p_*(\boldsymbol{x})}}\nabla\ln\frac{p_t(\boldsymbol{x})}{p_*(\boldsymbol{x})} - \int_{\mathbb{R}^d}\overline{k}_\sigma(\boldsymbol{x},\boldsymbol{x}-\boldsymbol{y})\frac{p_t(\boldsymbol{x}-\boldsymbol{y})}{\sqrt{p_*(\boldsymbol{x}-\boldsymbol{y})}}\nabla\ln\frac{p_t(\boldsymbol{x}-\boldsymbol{y})}{\sqrt{p_*(\boldsymbol{x}-\boldsymbol{y})}}d\boldsymbol{y}\right\|^2 d\boldsymbol{x}$$

$$\leq 2\int_{\mathbb{R}^d}\left\|\int_{\mathbb{R}^d}\overline{k}_\sigma(\boldsymbol{x},\boldsymbol{x}-\boldsymbol{y})\left[\frac{p_t(\boldsymbol{x})}{\sqrt{p_*(\boldsymbol{x})}}\nabla\ln\frac{p_t(\boldsymbol{x})}{p_*(\boldsymbol{x})} - \frac{p_t(\boldsymbol{x}-\boldsymbol{y})}{\sqrt{p_*(\boldsymbol{x}-\boldsymbol{y})}}\nabla\ln\frac{p_t(\boldsymbol{x}-\boldsymbol{y})}{p_*(\boldsymbol{x}-\boldsymbol{y})}\right]d\boldsymbol{y}\right\|^2 d\boldsymbol{x}$$

$$+ 2\int_{\mathbb{R}^d}\left(1 - \int_{\mathbb{R}^d}\overline{k}_\sigma(\boldsymbol{x},\boldsymbol{x}-\boldsymbol{y})d\boldsymbol{y}\right)^2\left\|\frac{p_t(\boldsymbol{x})}{\sqrt{p_*(\boldsymbol{x})}}\nabla\ln\frac{p_t(\boldsymbol{x})}{p_*(\boldsymbol{x})}\right\|^2 d\boldsymbol{x}. \tag{47}$$

where the first equation follows from the change of variable. With the requirement of $\overline{k}$, we have

$$\overline{k}_\sigma(\boldsymbol{x},\boldsymbol{y}) = \tilde{k}_\sigma(\boldsymbol{x}-\boldsymbol{y}) = \frac{1}{\sigma}\tilde{k}\left(\frac{(\boldsymbol{x}-\boldsymbol{y})}{\sigma}\right), \quad \int_{\mathbb{R}^d}\|\boldsymbol{y}\|^4\cdot\tilde{k}(\boldsymbol{y})d\boldsymbol{y} \leq M$$

$$\text{and} \quad \left|1 - \int_{\mathbb{R}^d}\overline{k}_\sigma(\boldsymbol{x},\boldsymbol{x}-\boldsymbol{y})d\boldsymbol{y}\right| \leq \frac{1}{2\sqrt{2}}. \tag{48}$$

In this condition, suppose $\boldsymbol{y} = \sigma\boldsymbol{z}$ in Eq. 47, $\boldsymbol{x_z} := \boldsymbol{x} - \sigma\boldsymbol{z}$, then we have

$$\text{KAE} \leq 4\underbrace{\int_{\mathbb{R}^d}\left\|\int_{\mathbb{R}^d}\tilde{k}(\boldsymbol{z})\left[\frac{p_t(\boldsymbol{x})}{\sqrt{p_*(\boldsymbol{x})}}\nabla\ln\frac{p_t(\boldsymbol{x})}{p_*(\boldsymbol{x})} - \frac{p_t(\boldsymbol{x_z})}{\sqrt{p_*(\boldsymbol{x_z})}}\nabla\ln\frac{p_t(\boldsymbol{x_z})}{p_*(\boldsymbol{x_z})}\right]d\boldsymbol{z}\right\|^2 d\boldsymbol{x}}_{\text{Term 1}}$$

$$+ \frac{1}{4}\int_{\mathbb{R}^d}p_*(\boldsymbol{x})\cdot\left\|\frac{p_t(\boldsymbol{x})}{p_*(\boldsymbol{x})}\nabla\ln\frac{p_t(\boldsymbol{x})}{p_*(\boldsymbol{x})}\right\|^2 d\boldsymbol{x}. \tag{49}$$

For any $\boldsymbol{x} \in \mathbb{R}^d$, we have

$$\frac{p_t(\boldsymbol{x})}{\sqrt{p_*(\boldsymbol{x})}}\nabla\ln\frac{p_t(\boldsymbol{x})}{p_*(\boldsymbol{x})} = \frac{p_t(\boldsymbol{x})}{\sqrt{p_*(\boldsymbol{x})}}\left(\nabla f_*(\boldsymbol{x}) - \nabla f_t(\boldsymbol{x})\right) \quad \text{where} \quad p_t(\boldsymbol{x}) = e^{-f_t(\boldsymbol{x})}, p_*(\boldsymbol{x}) = e^{-f_*(\boldsymbol{x})}.$$

Plugging such an equation into Eq. 49, we have

$$
\begin{aligned}
\text{Term 1} =& \int_{\mathbb{R}^d} \left\| \int_{\mathbb{R}^d} \tilde{k}(\boldsymbol{z}) \left[ \left( \frac{p_t(\boldsymbol{x})}{\sqrt{p_*(\boldsymbol{x})}} \nabla f_*(\boldsymbol{x}) - \frac{p_t(\boldsymbol{x_z})}{\sqrt{p_*(\boldsymbol{x_z})}} \nabla f_*(\boldsymbol{x_z}) \right) \right. \right. \\
& \left. \left. - \left( \frac{p_t(\boldsymbol{x})}{\sqrt{p_*(\boldsymbol{x})}} \nabla f_t(\boldsymbol{x}) - \frac{p_t(\boldsymbol{x_z})}{\sqrt{p_*(\boldsymbol{x_z})}} \nabla f_t(\boldsymbol{x_z}) \right) \right] \mathrm{d}\boldsymbol{z} \right\|^2 \mathrm{d}\boldsymbol{x} \\
\leq& 2 \int_{\mathbb{R}^d} \left\| \int_{\mathbb{R}^d} \tilde{k}(\boldsymbol{z}) \left( \frac{p_t(\boldsymbol{x})}{\sqrt{p_*(\boldsymbol{x})}} \nabla f_*(\boldsymbol{x}) - \frac{p_t(\boldsymbol{x_z})}{\sqrt{p_*(\boldsymbol{x_z})}} \nabla f_*(\boldsymbol{x_z}) \right) \mathrm{d}\boldsymbol{z} \right\|^2 \mathrm{d}\boldsymbol{x} \\
& + 2 \int_{\mathbb{R}^d} \left\| \int_{\mathbb{R}^d} \tilde{k}(\boldsymbol{z}) \left( \frac{p_t(\boldsymbol{x})}{\sqrt{p_*(\boldsymbol{x})}} \nabla f_t(\boldsymbol{x}) - \frac{p_t(\boldsymbol{x_z})}{\sqrt{p_*(\boldsymbol{x_z})}} \nabla f_t(\boldsymbol{x_z}) \right) \mathrm{d}\boldsymbol{z} \right\|^2 \mathrm{d}\boldsymbol{x} \\
\leq& 2 \int_{\mathbb{R}^d} \left( \int_{\mathbb{R}^d} \tilde{k}(\boldsymbol{z}) \mathrm{d}\boldsymbol{z} \cdot \int_{\mathbb{R}^d} \tilde{k}(\boldsymbol{z}) \left\| \left( \frac{p_t(\boldsymbol{x})}{\sqrt{p_*(\boldsymbol{x})}} \nabla f_*(\boldsymbol{x}) - \frac{p_t(\boldsymbol{x_z})}{\sqrt{p_*(\boldsymbol{x_z})}} \nabla f_*(\boldsymbol{x_z}) \right) \right\|^2 \mathrm{d}\boldsymbol{z} \right) \mathrm{d}\boldsymbol{x} \\
& + 2 \int_{\mathbb{R}^d} \left( \int_{\mathbb{R}^d} \tilde{k}(\boldsymbol{z}) \mathrm{d}\boldsymbol{z} \cdot \int_{\mathbb{R}^d} \tilde{k}(\boldsymbol{z}) \left\| \left( \frac{p_t(\boldsymbol{x})}{\sqrt{p_*(\boldsymbol{x})}} \nabla f_t(\boldsymbol{x}) - \frac{p_t(\boldsymbol{x_z})}{\sqrt{p_*(\boldsymbol{x_z})}} \nabla f_t(\boldsymbol{x_z}) \right) \right\|^2 \mathrm{d}\boldsymbol{z} \right) \mathrm{d}\boldsymbol{x} \\
\leq& \underbrace{3 \int_{\mathbb{R}^d} \int_{\mathbb{R}^d} \tilde{k}(\boldsymbol{z}) \left\| \frac{p_t(\boldsymbol{x})}{\sqrt{p_*(\boldsymbol{x})}} \nabla f_t(\boldsymbol{x}) - \frac{p_t(\boldsymbol{x_z})}{\sqrt{p_*(\boldsymbol{x_z})}} \nabla f_t(\boldsymbol{x_z}) \right\|^2 \mathrm{d}\boldsymbol{z}\mathrm{d}\boldsymbol{x}}_{\text{Term 1.1}} \\
& + \underbrace{3 \int_{\mathbb{R}^d} \int_{\mathbb{R}^d} \tilde{k}(\boldsymbol{z}) \left\| \frac{p_t(\boldsymbol{x})}{\sqrt{p_*(\boldsymbol{x})}} \nabla f_*(\boldsymbol{x}) - \frac{p_t(\boldsymbol{x_z})}{\sqrt{p_*(\boldsymbol{x_z})}} \nabla f_*(\boldsymbol{x_z}) \right\|^2 \mathrm{d}\boldsymbol{z}\mathrm{d}\boldsymbol{x}}_{\text{Term 1.2}}
\end{aligned}
\tag{50}
$$

where the first inequality follows from Minkowski inequality, the second inequality follows from Cauchy–Schwarz inequality, and the third inequality follows from Eq. 48.

Consider Term 1.1, we have,

$$
\begin{aligned}
\text{Term 1.1} =& \int_{\mathbb{R}^d} \int_{\mathbb{R}^d} \tilde{k}(\boldsymbol{z}) \left\| \frac{p_t(\boldsymbol{x})}{\sqrt{p_*(\boldsymbol{x})}} \nabla f_t(\boldsymbol{x}) - \frac{p_t(\boldsymbol{x_z})}{\sqrt{p_*(\boldsymbol{x_z})}} \nabla f_t(\boldsymbol{x}) \right. \\
& \left. + \frac{p_t(\boldsymbol{x_z})}{\sqrt{p_*(\boldsymbol{x_z})}} \nabla f_t(\boldsymbol{x}) - \frac{p_t(\boldsymbol{x_z})}{\sqrt{p_*(\boldsymbol{x_z})}} \nabla f_t(\boldsymbol{x_z}) \right\|^2 \mathrm{d}\boldsymbol{z}\mathrm{d}\boldsymbol{x} \\
\leq& 2 \int_{\mathbb{R}^d} \int_{\mathbb{R}^d} \tilde{k}(\boldsymbol{z}) \left\| \left( \frac{p_t(\boldsymbol{x})}{\sqrt{p_*(\boldsymbol{x})}} - \frac{p_t(\boldsymbol{x_z})}{\sqrt{p_*(\boldsymbol{x_z})}} \right) \nabla f_t(\boldsymbol{x}) \right\|^2 \mathrm{d}\boldsymbol{z}\mathrm{d}\boldsymbol{x} \\
& + 2 \int_{\mathbb{R}^d} \int_{\mathbb{R}^d} \tilde{k}(\boldsymbol{z}) \left\| \frac{p_t(\boldsymbol{x_z})}{\sqrt{p_*(\boldsymbol{x_z})}} \left( \nabla f_t(\boldsymbol{x}) - \nabla f_t(\boldsymbol{x_z}) \right) \right\|^2 \mathrm{d}\boldsymbol{z}\mathrm{d}\boldsymbol{x} \\
\leq& 2 \int_{\mathbb{R}^d} \int_{\mathbb{R}^d} \tilde{k}(\boldsymbol{z}) \left\| \left( \frac{p_t(\boldsymbol{x})}{\sqrt{p_*(\boldsymbol{x})}} - \frac{p_t(\boldsymbol{x_z})}{\sqrt{p_*(\boldsymbol{x_z})}} \right) \nabla f_t(\boldsymbol{x}) \right\|^2 \mathrm{d}\boldsymbol{z}\mathrm{d}\boldsymbol{x} \\
& + 2 \int_{\mathbb{R}^d} \tilde{k}(\boldsymbol{z}) L^2 \sigma^2 \|\boldsymbol{z}\|^2 \int_{\mathbb{R}^d} \left( \frac{p_t(\boldsymbol{x_z})}{\sqrt{p_*(\boldsymbol{x_z})}} \right)^2 \mathrm{d}\boldsymbol{x}\mathrm{d}\boldsymbol{z} \\
\leq& 2 \int_{\mathbb{R}^d} \int_{\mathbb{R}^d} \tilde{k}(\boldsymbol{z}) \left\| \left( \frac{p_t(\boldsymbol{x})}{\sqrt{p_*(\boldsymbol{x})}} - \frac{p_t(\boldsymbol{x_z})}{\sqrt{p_*(\boldsymbol{x_z})}} \right) \nabla f_t(\boldsymbol{x}) \right\|^2 \mathrm{d}\boldsymbol{z}\mathrm{d}\boldsymbol{x} + 2\sigma^2 \beta L^2 (M+1),
\end{aligned}
\tag{51}
$$

where the second inequality follows from $L$-smoothness of $p_t(\boldsymbol{x})$ and the Fubini's theorem, and the third inequality follows from $p_t$ warm (Assumption 3) and the fact $\int_{\mathbb{R}^d} \|\boldsymbol{y}\|^2 \cdot \tilde{k}(\boldsymbol{y}) \mathrm{d}\boldsymbol{y} \leq D$ in Eq. 48.

In the following, we focus on the first term of RHS of Eq. 51, and have

$$\frac{p_t(\boldsymbol{x})}{\sqrt{p_*(\boldsymbol{x})}} = \exp\left(\frac{f_*(\boldsymbol{x})}{2} - f_t(\boldsymbol{x})\right). \tag{52}$$

For each $\boldsymbol{x} \in \mathbb{R}^d$, suppose high dimensional $\mathbb{R}^d$ can be divided into two parts:

$$\begin{aligned} \mathcal{B}_l(\boldsymbol{x}) &= \left\{ \boldsymbol{z} \,\middle|\, \frac{f_*(\boldsymbol{x})}{2} - f_t(\boldsymbol{x}) \leq \frac{f_*(\boldsymbol{x_z})}{2} - f_t(\boldsymbol{x_z}) \right\}, \\ \mathcal{B}_u(\boldsymbol{x}) &= \left\{ \boldsymbol{z} \,\middle|\, \frac{f_*(\boldsymbol{x})}{2} - f_t(\boldsymbol{x}) \geq \frac{f_*(\boldsymbol{x_z})}{2} - f_t(\boldsymbol{x_z}) \right\}. \end{aligned} \tag{53}$$

For $\boldsymbol{z} \in \mathcal{B}_l(\boldsymbol{x})$, we have

$$\begin{aligned} &\frac{p_t(\boldsymbol{x_z})}{\sqrt{p_*(\boldsymbol{x_z})}} - \frac{p_t(\boldsymbol{x})}{\sqrt{p_*(\boldsymbol{x})}} = \frac{p_t(\boldsymbol{x_z})}{\sqrt{p_*(\boldsymbol{x_z})}} \cdot \left(1 - \exp\left(\left(\frac{f_*(\boldsymbol{x})}{2} - f_t(\boldsymbol{x})\right) - \left(\frac{f_*(\boldsymbol{x_z})}{2} - f_t(\boldsymbol{x_z})\right)\right)\right) \\ \leq& \frac{p_t(\boldsymbol{x_z})}{\sqrt{p_*(\boldsymbol{x_z})}} \cdot \left[\left(\frac{f_*(\boldsymbol{x_z})}{2} - f_t(\boldsymbol{x_z})\right) - \left(\frac{f_*(\boldsymbol{x})}{2} - f_t(\boldsymbol{x})\right)\right] \\ \leq& \frac{p_t(\boldsymbol{x_z})}{\sqrt{p_*(\boldsymbol{x_z})}} \cdot \left[-\frac{\sigma}{2}\left(\nabla f_*(\boldsymbol{x}) - \nabla f_*(\boldsymbol{x_z}) + f_*(\boldsymbol{x} - \sigma \boldsymbol{z})\right)^\top \boldsymbol{z} + \sigma \nabla f_t(\boldsymbol{x_z})^\top \boldsymbol{z} + \frac{3L\sigma^2}{4}\|\boldsymbol{z}\|^2\right] \\ \leq& \frac{p_t(\boldsymbol{x_z})}{\sqrt{p_*(\boldsymbol{x_z})}} \cdot \left[-\frac{\sigma}{2}\nabla f_*(\boldsymbol{x_z})^\top \boldsymbol{z} + \sigma \nabla f_t(\boldsymbol{x_z})^\top \boldsymbol{z} + \frac{5L\sigma^2}{4}\|\boldsymbol{z}\|^2\right], \end{aligned} \tag{54}$$

where the first inequality follows from $1 - e^{-x} \leq x$ for any $x \geq -1$, the second and third inequality follows from L-smoothness of $f_*$ and $f_t$. Therefore, we have

$$\begin{aligned} &\int_{\mathbb{R}^d} \int_{\mathcal{B}_l} \tilde{k}(\boldsymbol{z}) \left\|\left(\frac{p_t(\boldsymbol{x})}{\sqrt{p_*(\boldsymbol{x})}} - \frac{p_t(\boldsymbol{x_z})}{\sqrt{p_*(\boldsymbol{x_z})}}\right) \nabla f_t(\boldsymbol{x})\right\|^2 \mathrm{d}\boldsymbol{z}\mathrm{d}\boldsymbol{x} \\ \leq& \int_{\mathbb{R}^d} \int_{\mathcal{B}_l} \tilde{k}(\boldsymbol{z}) \left\|\frac{p_t(\boldsymbol{x_z})}{\sqrt{p_*(\boldsymbol{x_z})}} \cdot \left[-\frac{\sigma}{2}\nabla f_*(\boldsymbol{x_z})^\top \boldsymbol{z} + \sigma \nabla f_t(\boldsymbol{x_z})^\top \boldsymbol{z} + \frac{5L\sigma^2}{4}\|\boldsymbol{z}\|^2\right] \cdot \nabla f_t(\boldsymbol{x})\right\|^2 \mathrm{d}\boldsymbol{z}\mathrm{d}\boldsymbol{x} \\ \leq& \int_{\mathbb{R}^d} \int_{\mathcal{B}_l} \frac{\tilde{k}(\boldsymbol{z})}{2\sigma} \left[-\frac{\sigma}{2}\nabla f_*(\boldsymbol{x_z})^\top \boldsymbol{z} + \sigma \nabla f_t(\boldsymbol{x_z})^\top \boldsymbol{z} + \frac{5L\sigma^2}{4}\|\boldsymbol{z}\|^2\right]^2 \sqrt{p_t(\boldsymbol{x_z})}\mathrm{d}\boldsymbol{z}\mathrm{d}\boldsymbol{x} \\ &+ \int_{\mathbb{R}^d} \int_{\mathcal{B}_l} \frac{\tilde{k}(\boldsymbol{z})\sigma}{2} \cdot \frac{(p_t(\boldsymbol{x_z}))^{1.5}}{p_*(\boldsymbol{x_z})} \cdot \|\nabla f_t(\boldsymbol{x}) - \nabla f_t(\boldsymbol{x_z}) + \nabla f_t(\boldsymbol{x_z})\|^2 \mathrm{d}\boldsymbol{z}\mathrm{d}\boldsymbol{x} \\ \leq& \int_{\mathbb{R}^d} \int_{\mathcal{B}_l} \sigma \cdot \tilde{k}(\boldsymbol{z}) \|\boldsymbol{z}\|^2 \sqrt{p_t(\boldsymbol{x_z})} \cdot \left(\frac{1}{2}\|\nabla f_*(\boldsymbol{x_z})\|^2 + \frac{3}{2}\|\nabla f_t(\boldsymbol{x_z})\|^2 + \frac{5L^2\sigma^2}{2}\|\boldsymbol{z}\|^2\right) \mathrm{d}\boldsymbol{z}\mathrm{d}\boldsymbol{x} \\ &+ \int_{\mathbb{R}^d} \int_{\mathcal{B}_l} \sigma \cdot \tilde{k}(\boldsymbol{z}) \cdot \frac{(p_t(\boldsymbol{x_z}))^{1.5}}{p_*(\boldsymbol{x_z})} \cdot \left(\|\nabla f_t(\boldsymbol{x_z})\|^2 + L^2\sigma^2\|\boldsymbol{z}\|^2\right) \mathrm{d}\boldsymbol{z}\mathrm{d}\boldsymbol{x} \\ \leq& \frac{\sqrt{\beta}\sigma}{2} \cdot \int_{\mathbb{R}^d} \tilde{k}(\boldsymbol{z})\|\boldsymbol{z}\|^2 \int_{\mathbb{R}^d} \sqrt{p_*(\boldsymbol{x_z})} \|\nabla f_*(\boldsymbol{x_z})\|^2 \mathrm{d}\boldsymbol{x}\mathrm{d}\boldsymbol{z} \\ &+ \frac{3\sigma}{2} \cdot \int_{\mathbb{R}^d} \tilde{k}(\boldsymbol{z})\|\boldsymbol{z}\|^2 \int_{\mathbb{R}^d} \sqrt{p_t(\boldsymbol{x_z})} \|\nabla f_t(\boldsymbol{x_z})\|^2 \mathrm{d}\boldsymbol{x}\mathrm{d}\boldsymbol{z} + \frac{5L^2\sigma^3}{2} \cdot \int_{\mathbb{R}^d} \tilde{k}(\boldsymbol{z})\|\boldsymbol{z}\|^4 \int_{\mathbb{R}^d} \sqrt{p_t(\boldsymbol{x_z})}\mathrm{d}\boldsymbol{x}\mathrm{d}\boldsymbol{z} \\ &+ \beta\sigma \cdot \int_{\mathbb{R}^d} \tilde{k}(\boldsymbol{z}) \int_{\mathbb{R}^d} \sqrt{p_t(\boldsymbol{x_z})} \|\nabla f_t(\boldsymbol{x_z})\|^2 \mathrm{d}\boldsymbol{x}\mathrm{d}\boldsymbol{z} + \beta L^2\sigma^3 \cdot \int_{\mathbb{R}^d} \tilde{k}(\boldsymbol{z})\|\boldsymbol{z}\|^2 \int_{\mathbb{R}^d} \sqrt{p_t(\boldsymbol{x_z})}\mathrm{d}\boldsymbol{x}\mathrm{d}\boldsymbol{z} \\ \leq& \sqrt{\beta}CdL(M+1) \cdot \sigma + 3\sqrt{\beta}CdL(M+1) \cdot \sigma + \frac{5}{4}\sqrt{\beta}CL^2M \cdot \sigma^3 + 3\beta^{1.5}CdL \cdot \sigma + \frac{1}{2}\beta^{1.5}CL^2(M+1) \cdot \sigma^3. \end{aligned} \tag{55}$$

It can be noticed that the first inequality follows from Eq. 54, the second and the third inequalities follow from Cauchy–Schwarz inequality, the fourth inequality follows from the $\beta$-warm during

the update, the fourth inequality follows from the $\beta$-warm during the update, Lemma B.2 and the following fact

$$\int_{\mathcal{B}_l} \tilde{k}(\boldsymbol{z}) \mathrm{d}\boldsymbol{z} \leq \int_{\mathbb{R}^d} \tilde{k}(\boldsymbol{z}) \mathrm{d}\boldsymbol{z}.$$

Besides, the constant $C$ is provided by Lemma B.1 as

$$\int_{\mathbb{R}^d} \sqrt{p_*(\boldsymbol{x})} \mathrm{d}\boldsymbol{x} \leq C.$$

For $\boldsymbol{z} \in \mathcal{B}_u(\boldsymbol{x})$, we have

$$
\begin{aligned}
&\frac{p_t(\boldsymbol{x})}{\sqrt{p_*(\boldsymbol{x})}} - \frac{p_t(\boldsymbol{x_z})}{\sqrt{p_*(\boldsymbol{x_z})}} \\
=& \frac{p_t(\boldsymbol{x})}{\sqrt{p_*(\boldsymbol{x})}} \cdot \left( 1 - \exp\left( \left( \frac{f_*(\boldsymbol{x_z})}{2} - f_t(\boldsymbol{x_z}) \right) - \left( \frac{f_*(\boldsymbol{x})}{2} - f_t(\boldsymbol{x}) \right) \right) \right) \\
\leq& \frac{p_t(\boldsymbol{x})}{\sqrt{p_*(\boldsymbol{x})}} \cdot \left[ \left( \frac{f_*(\boldsymbol{x})}{2} - f_t(\boldsymbol{x}) \right) - \left( \frac{f_*(\boldsymbol{x_z})}{2} - f_t(\boldsymbol{x_z}) \right) \right] \\
\leq& \frac{p_t(\boldsymbol{x})}{\sqrt{p_*(\boldsymbol{x})}} \cdot \left[ \frac{\sigma}{2} \left( \nabla f_*(\boldsymbol{x_z}) - \nabla f_*(\boldsymbol{x}) + \nabla f_*(\boldsymbol{x}) \right)^\top \boldsymbol{z} - \sigma \nabla f_t^\top(\boldsymbol{x}) \boldsymbol{z} + \frac{3L\sigma^2}{4} \|\boldsymbol{z}\|^2 \right] \\
\leq& \frac{p_t(\boldsymbol{x})}{\sqrt{p_*(\boldsymbol{x})}} \cdot \left[ \frac{\sigma}{2} \nabla f_*^\top(\boldsymbol{x}) \boldsymbol{z} - \sigma \nabla f_t^\top(\boldsymbol{x}) \boldsymbol{z} + \frac{5L\sigma^2}{4} \|\boldsymbol{z}\|^2 \right].
\end{aligned}
\tag{56}
$$

Similar to Eq. 55, we have

$$
\begin{aligned}
& \int_{\mathbb{R}^d} \int_{\mathcal{B}_u} \tilde{k}(\boldsymbol{z}) \left\| \left( \frac{p_t(\boldsymbol{x})}{\sqrt{p_*(\boldsymbol{x})}} - \frac{p_t(\boldsymbol{x_z})}{\sqrt{p_*(\boldsymbol{x_z})}} \right) \nabla f_t(\boldsymbol{x}) \right\|^2 \mathrm{d}\boldsymbol{z}\mathrm{d}\boldsymbol{x} \\
\leq& \int_{\mathbb{R}^d} \int_{\mathcal{B}_u} \tilde{k}(\boldsymbol{z}) \left\| \frac{p_t(\boldsymbol{x})}{\sqrt{p_*(\boldsymbol{x})}} \cdot \left[ \frac{\sigma}{2} \nabla f_*^\top(\boldsymbol{x}) \boldsymbol{z} - \sigma \nabla f_t^\top(\boldsymbol{x}) \boldsymbol{z} + \frac{5L\sigma^2}{4} \|\boldsymbol{z}\|^2 \right] \cdot \nabla f_t(\boldsymbol{x}) \right\|^2 \mathrm{d}\boldsymbol{z}\mathrm{d}\boldsymbol{x} \\
\leq& \int_{\mathbb{R}^d} \int_{\mathcal{B}_u} \frac{\tilde{k}(\boldsymbol{z})}{2\sigma} \left[ \frac{\sigma}{2} \nabla f_*^\top(\boldsymbol{x}) \boldsymbol{z} - \sigma \nabla f_t^\top(\boldsymbol{x}) \boldsymbol{z} + \frac{5L\sigma^2}{4} \|\boldsymbol{z}\|^2 \right]^2 \cdot \sqrt{p_t(\boldsymbol{x})} \mathrm{d}\boldsymbol{z}\mathrm{d}\boldsymbol{x} \\
& + \int_{\mathbb{R}^d} \int_{\mathcal{B}_u} \frac{\tilde{k}(\boldsymbol{z})\sigma}{2} \cdot \frac{(p_t(\boldsymbol{x}))^{1.5}}{p_*(\boldsymbol{x})} \cdot \|\nabla f_t(\boldsymbol{x})\|^2 \mathrm{d}\boldsymbol{z}\mathrm{d}\boldsymbol{x} \\
\leq& \int_{\mathbb{R}^d} \int_{\mathcal{B}_u} \sigma \cdot \tilde{k}(\boldsymbol{z}) \|\boldsymbol{z}\|^2 \sqrt{p_t(\boldsymbol{x})} \cdot \left( \frac{1}{2} \|\nabla f_*(\boldsymbol{x})\|^2 + \frac{3}{2} \|\nabla f_t(\boldsymbol{x})\|^2 + \frac{5L^2\sigma^2}{2} \|\boldsymbol{z}\|^2 \right) \mathrm{d}\boldsymbol{z}\mathrm{d}\boldsymbol{x} \\
& + \int_{\mathbb{R}^d} \int_{\mathcal{B}_u} \frac{\tilde{k}(\boldsymbol{z})\sigma}{2} \cdot \frac{(p_t(\boldsymbol{x}))^{1.5}}{p_*(\boldsymbol{x})} \cdot \|\nabla f_t(\boldsymbol{x})\|^2 \mathrm{d}\boldsymbol{z}\mathrm{d}\boldsymbol{x} \\
\leq& \frac{\sqrt{\beta}\sigma}{2} \cdot \int_{\mathbb{R}^d} \tilde{k}(\boldsymbol{z})\|\boldsymbol{z}\|^2 \int_{\mathbb{R}^d} \sqrt{p_*(\boldsymbol{x})} \|\nabla f_*(\boldsymbol{x})\|^2 \mathrm{d}\boldsymbol{x}\mathrm{d}\boldsymbol{z} + \frac{3\sigma}{2} \cdot \int_{\mathbb{R}^d} \tilde{k}(\boldsymbol{z})\|\boldsymbol{z}\|^2 \int_{\mathbb{R}^d} \sqrt{p_t(\boldsymbol{x})} \|\nabla f_t(\boldsymbol{x})\|^2 \mathrm{d}\boldsymbol{x}\mathrm{d}\boldsymbol{z} \\
& + \frac{5L^2\sigma^3}{2} \cdot \int_{\mathbb{R}^d} \tilde{k}(\boldsymbol{z})\|\boldsymbol{z}\|^4 \int_{\mathbb{R}^d} \sqrt{p_t(\boldsymbol{x})} \mathrm{d}\boldsymbol{x}\mathrm{d}\boldsymbol{z} + \frac{\beta\sigma}{2} \cdot \int_{\mathbb{R}^d} \tilde{k}(\boldsymbol{z}) \int_{\mathbb{R}^d} \sqrt{p_t(\boldsymbol{x})} \|\nabla f_t(\boldsymbol{x})\|^2 \mathrm{d}\boldsymbol{x}\mathrm{d}\boldsymbol{z} \\
\leq& \sqrt{\beta}CdL(M+1) \cdot \sigma + 3\sqrt{\beta}CdL(M+1) \cdot \sigma + \frac{5}{4}\sqrt{\beta}CL^2M \cdot \sigma^3 + \frac{3}{2}\beta^{1.5}CdL \cdot \sigma.
\end{aligned}
\tag{57}
$$

Plugging Eq. 55, Eq. 57 into Eq. 51, we have

$$
\begin{aligned}
\text{Term 1.1} \leq& 16\sqrt{\beta}CdL(M+1) \cdot \sigma + 9\beta^{1.5}CdL \cdot \sigma + 2\beta L^2(M+1) \cdot \sigma^2 \\
& + 5\sqrt{\beta}CL^2M \cdot \sigma^3 + \beta^{1.5}CL^2(M+1) \cdot \sigma^3.
\end{aligned}
\tag{58}
$$

With the same techniques in Eq. 51, we have

$$
\text{Term 1.2} \leq 2 \int_{\mathbb{R}^d} \int_{\mathbb{B}(\boldsymbol{0},r)} \tilde{k}(\boldsymbol{z}) \left\| \left( \frac{p_t(\boldsymbol{x})}{\sqrt{p_*(\boldsymbol{x})}} - \frac{p_t(\boldsymbol{x_z})}{\sqrt{p_*(\boldsymbol{x_z})}} \right) \nabla f_*(\boldsymbol{x}) \right\|^2 \mathrm{d}\boldsymbol{z}\mathrm{d}\boldsymbol{x} + 5L^2\sigma^2r^2.
\tag{59}
$$

Similar to Eq. 55, when $z \in \mathcal{B}_l(\boldsymbol{x})$, we have

$$
\begin{aligned}
&\int_{\mathbb{R}^d} \int_{\mathcal{B}_l} \tilde{k}(\boldsymbol{z}) \left\| \left( \frac{p_t(\boldsymbol{x})}{\sqrt{p_*(\boldsymbol{x})}} - \frac{p_t(\boldsymbol{x_z})}{\sqrt{p_*(\boldsymbol{x_z})}} \right) \nabla f_*(\boldsymbol{x}) \right\|^2 \mathrm{d}\boldsymbol{z}\mathrm{d}\boldsymbol{x} \\
\leq & \int_{\mathbb{R}^d} \int_{\mathcal{B}_l} \sigma \cdot \tilde{k}(\boldsymbol{z}) \|\boldsymbol{z}\|^2 \sqrt{p_t(\boldsymbol{x_z})} \cdot \left( \frac{1}{2} \|\nabla f_*(\boldsymbol{x_z})\|^2 + \frac{3}{2} \|\nabla f_t(\boldsymbol{x_z})\|^2 + \frac{5L^2\sigma^2}{2} \|\boldsymbol{z}\|^2 \right) \mathrm{d}\boldsymbol{z}\mathrm{d}\boldsymbol{x} \\
& + \int_{\mathbb{R}^d} \int_{\mathcal{B}_l} \sigma \cdot \tilde{k}(\boldsymbol{z}) \cdot \frac{(p_t(\boldsymbol{x_z}))^{1.5}}{p_*(\boldsymbol{x_z})} \cdot \left( \|\nabla f_*(\boldsymbol{x_z})\|^2 + L^2\sigma^2 \|\boldsymbol{z}\|^2 \right) \mathrm{d}\boldsymbol{z}\mathrm{d}\boldsymbol{x} \\
\leq & \frac{\sqrt{\beta}\sigma}{2} \cdot \int_{\mathbb{R}^d} \tilde{k}(\boldsymbol{z})\|\boldsymbol{z}\|^2 \int_{\mathbb{R}^d} \sqrt{p_*(\boldsymbol{x_z})} \|\nabla f_*(\boldsymbol{x_z})\|^2 \mathrm{d}\boldsymbol{x}\mathrm{d}\boldsymbol{z} \\
& + \frac{3\sigma}{2} \cdot \int_{\mathbb{R}^d} \tilde{k}(\boldsymbol{z})\|\boldsymbol{z}\|^2 \int_{\mathbb{R}^d} \sqrt{p_t(\boldsymbol{x_z})} \|\nabla f_t(\boldsymbol{x_z})\|^2 \mathrm{d}\boldsymbol{x}\mathrm{d}\boldsymbol{z} + \frac{5L^2\sigma^3}{2} \cdot \int_{\mathbb{R}^d} \tilde{k}(\boldsymbol{z})\|\boldsymbol{z}\|^4 \int_{\mathbb{R}^d} \sqrt{p_t(\boldsymbol{x_z})}\mathrm{d}\boldsymbol{x}\mathrm{d}\boldsymbol{z} \\
& + \beta\sigma \cdot \int_{\mathbb{R}^d} \tilde{k}(\boldsymbol{z}) \int_{\mathbb{R}^d} \sqrt{p_t(\boldsymbol{x_z})} \|\nabla f_*(\boldsymbol{x_z})\|^2 \mathrm{d}\boldsymbol{x}\mathrm{d}\boldsymbol{z} + \beta L^2\sigma^3 \cdot \int_{\mathbb{R}^d} \tilde{k}(\boldsymbol{z})\|\boldsymbol{z}\|^2 \int_{\mathbb{R}^d} \sqrt{p_t(\boldsymbol{x_z})}\mathrm{d}\boldsymbol{x}\mathrm{d}\boldsymbol{z} \\
\leq & \sqrt{\beta}CdL(M+1) \cdot \sigma + 3\sqrt{\beta}CdL(M+1) \cdot \sigma + \frac{5}{4}\sqrt{\beta}CL^2M \cdot \sigma^3 + 3\beta^{1.5}CdL \cdot \sigma + \frac{1}{2}\beta^{1.5}CL^2(M+1) \cdot \sigma^3,
\end{aligned}
\tag{60}
$$

where the last inequality utilizes additional $\beta$-warm condition comparing with Eq. 55.

Similar to Eq. 57, when $z \in \mathcal{B}_u(\boldsymbol{x})$, we have

$$
\begin{aligned}
&\int_{\mathbb{R}^d} \int_{\mathcal{B}_u} \tilde{k}(\boldsymbol{z}) \left\| \left( \frac{p_t(\boldsymbol{x})}{\sqrt{p_*(\boldsymbol{x})}} - \frac{p_t(\boldsymbol{x_z})}{\sqrt{p_*(\boldsymbol{x_z})}} \right) \nabla f_*(\boldsymbol{x}) \right\|^2 \mathrm{d}\boldsymbol{z}\mathrm{d}\boldsymbol{x} \\
\leq & \int_{\mathbb{R}^d} \int_{\mathcal{B}_u} \sigma \cdot \tilde{k}(\boldsymbol{z}) \|\boldsymbol{z}\|^2 \sqrt{p_t(\boldsymbol{x})} \cdot \left( \frac{1}{2} \|\nabla f_*(\boldsymbol{x})\|^2 + \frac{3}{2} \|\nabla f_t(\boldsymbol{x})\|^2 + \frac{5L^2\sigma^2}{2} \|\boldsymbol{z}\|^2 \right) \mathrm{d}\boldsymbol{z}\mathrm{d}\boldsymbol{x} \\
& + \int_{\mathbb{R}^d} \int_{\mathcal{B}_u} \frac{\tilde{k}(\boldsymbol{z})\sigma}{2} \cdot \frac{(p_t(\boldsymbol{x}))^{1.5}}{p_*(\boldsymbol{x})} \cdot \|\nabla f_*(\boldsymbol{x})\|^2 \mathrm{d}\boldsymbol{z}\mathrm{d}\boldsymbol{x} \\
\leq & \frac{\sqrt{\beta}\sigma}{2} \cdot \int_{\mathbb{R}^d} \tilde{k}(\boldsymbol{z})\|\boldsymbol{z}\|^2 \int_{\mathbb{R}^d} \sqrt{p_*(\boldsymbol{x})} \|\nabla f_*(\boldsymbol{x})\|^2 \mathrm{d}\boldsymbol{x}\mathrm{d}\boldsymbol{z} + \frac{3\sigma}{2} \cdot \int_{\mathbb{R}^d} \tilde{k}(\boldsymbol{z})\|\boldsymbol{z}\|^2 \int_{\mathbb{R}^d} \sqrt{p_t(\boldsymbol{x})} \|\nabla f_t(\boldsymbol{x})\|^2 \mathrm{d}\boldsymbol{x}\mathrm{d}\boldsymbol{z} \\
& + \frac{5L^2\sigma^3}{2} \cdot \int_{\mathbb{R}^d} \tilde{k}(\boldsymbol{z})\|\boldsymbol{z}\|^4 \int_{\mathbb{R}^d} \sqrt{p_t(\boldsymbol{x})}\mathrm{d}\boldsymbol{x}\mathrm{d}\boldsymbol{z} + \frac{\beta\sigma}{2} \cdot \int_{\mathbb{R}^d} \tilde{k}(\boldsymbol{z}) \int_{\mathbb{R}^d} \sqrt{p_t(\boldsymbol{x})} \|\nabla f_*(\boldsymbol{x})\|^2 \mathrm{d}\boldsymbol{x}\mathrm{d}\boldsymbol{z} \\
\leq & \sqrt{\beta}CdL(M+1) \cdot \sigma + 3\sqrt{\beta}CdL(M+1) \cdot \sigma + \frac{5}{4}\sqrt{\beta}CL^2M \cdot \sigma^3 + \frac{3}{2}\beta^{1.5}CdL \cdot \sigma.
\end{aligned}
\tag{61}
$$

Combining Eq. 60, Eq. 61 with Eq. 59, we have

$$
\begin{aligned}
\text{Term 1.2} \leq & 16\sqrt{\beta}CdL(M+1) \cdot \sigma + 9\beta^{1.5}CdL \cdot \sigma + 2\sigma^2\beta L^2(M+1) \\
& + 5\sqrt{\beta}CL^2M \cdot \sigma^3 + \beta^{1.5}CL^2(M+1) \cdot \sigma^3.
\end{aligned}
\tag{62}
$$

Without loss of generality, we suppose $\sigma \leq 1$ and $M \geq 1$. Plugging Eq. 58 and Eq. 62 into Eq. 50, we have

$$
\text{Term 1} \leq 12LM\sqrt{\beta}\sigma \cdot \left( 16Cd + \frac{9\beta Cd}{2M} + 6\sqrt{\beta}L + 3CL + \beta CL \right),
\tag{63}
$$

which means if we set

$$
\sigma = \min\left( 1, \frac{\epsilon}{12LM\sqrt{\beta}} \cdot \left( 16Cd + \frac{9\beta Cd}{2M} + 6\sqrt{\beta}L + 3CL + \beta CL \right)^{-1} \right),
\tag{64}
$$

KAE satisfies

$$
\text{KAE} \leq 4\epsilon + \frac{1}{4} \int_{\mathbb{R}^d} p_*(\boldsymbol{x}) \cdot \left\| \frac{p_t(\boldsymbol{x})}{p_*(\boldsymbol{x})} \nabla \ln \frac{p_t(\boldsymbol{x})}{p_*(\boldsymbol{x})} \right\|^2 \mathrm{d}\boldsymbol{x},
$$

and the proof is completed. $\qquad\square$

## C   THE MAIN THEOREM IN SECTION 3

Before providing the main theorem, i.e., Theorem 3.1, we need to introduce some lemmas.

**Lemma C.1.** *Suppose Assumption [$A_1$]–[$A_3$] are satisfied, and $p_0$ is near to the target $p_*$ satisfying $D_\chi(p_0, p_*) \leq 1/4$, for any time $T = -C \ln \epsilon$ where $\epsilon \leq (16\beta C)^{-2}$, we have*

$$D_\chi(p_T, p_*) \leq 1/2.$$

*Proof.* We denote $D_{\chi^2}(p_t, p_*)$ as chi-square distance between $p_t$ and $p_*$. We have the following functional derivative

$$\frac{\mathrm{d}D_\chi(p_t, p_*)}{\mathrm{d}t} = \int_{\mathbb{R}^d} \frac{\delta D_\chi(p_t, p_*)}{\delta p}(p_t)\partial_t p_t \mathrm{d}\boldsymbol{x} = \int_{\mathbb{R}^d} 2\frac{p_t(\boldsymbol{x})}{p_*(\boldsymbol{x})}\partial_t p_t(\boldsymbol{x})\mathrm{d}\boldsymbol{x}. \tag{65}$$

Combining the result with Remark 2, we have

$$\begin{aligned}
\frac{\mathrm{d}D_\chi(p_t, p_*)}{\mathrm{d}t} &= -\int_{\mathbb{R}^d} 2\nabla\frac{p_t(\boldsymbol{x})}{p_*(\boldsymbol{x})} \cdot \int_{\mathbb{R}^d} p_t(\boldsymbol{y})k(\boldsymbol{x}, \boldsymbol{y}) \cdot \nabla \ln\frac{p_t(\boldsymbol{y})}{p_*(\boldsymbol{y})}\mathrm{d}\boldsymbol{y}\mathrm{d}\boldsymbol{x} \\
&= -\int_{\mathbb{R}^d} 2\nabla\frac{p_t(\boldsymbol{x})}{p_*(\boldsymbol{x})} \cdot \int_{\mathbb{R}^d} p_*(\boldsymbol{y})k(\boldsymbol{x}, \boldsymbol{y}) \cdot \nabla\frac{p_t(\boldsymbol{y})}{p_*(\boldsymbol{y})}\mathrm{d}\boldsymbol{y}\mathrm{d}\boldsymbol{x}
\end{aligned} \tag{66}$$

Plugging Eq. 20 to the previous equation, we have

$$\begin{aligned}
\frac{\mathrm{d}D_\chi(p_t, p_*)}{\mathrm{d}t} = &-2\int_{\mathbb{R}^d} \frac{p_t(\boldsymbol{x})}{\sqrt{p_*(\boldsymbol{x})}} \cdot \nabla\frac{p_t(\boldsymbol{x})}{p_*(\boldsymbol{x})} \cdot \int_{\mathbb{R}^d} \overline{k}(\boldsymbol{x}, \boldsymbol{y})\sqrt{p_*(\boldsymbol{y})} \cdot \nabla\frac{p_t(\boldsymbol{y})}{p_*(\boldsymbol{y})}\mathrm{d}\boldsymbol{y}\mathrm{d}\boldsymbol{x} \\
= &-2\int_{\mathbb{R}^d} p_t(\boldsymbol{x})\left\|\nabla\frac{p_t(\boldsymbol{x})}{p_*(\boldsymbol{x})}\right\|^2 \mathrm{d}\boldsymbol{x} - 2\int_{\mathbb{R}^d} \frac{p_t(\boldsymbol{x})}{\sqrt{p_*(\boldsymbol{x})}} \cdot \nabla\frac{p_t(\boldsymbol{x})}{p_*(\boldsymbol{x})}\cdot \\
&\left(\int_{\mathbb{R}^d} \overline{k}(\boldsymbol{x}, \boldsymbol{y})\sqrt{p_*(\boldsymbol{y})} \cdot \nabla\frac{p_t(\boldsymbol{y})}{p_*(\boldsymbol{y})} - \sqrt{p_*(\boldsymbol{x})} \cdot \nabla\frac{p_t(\boldsymbol{x})}{p_*(\boldsymbol{x})}\right)\mathrm{d}\boldsymbol{x} \\
\leq &-\int_{\mathbb{R}^d} p_t(\boldsymbol{x})\left\|\nabla\frac{p_t(\boldsymbol{x})}{p_*(\boldsymbol{x})}\right\|^2 \mathrm{d}\boldsymbol{x} \\
&+\int_{\mathbb{R}^d} \frac{p_t(\boldsymbol{x})}{p_*(\boldsymbol{x})}\left\|\int_{\mathbb{R}^d} \overline{k}(\boldsymbol{x}, \boldsymbol{y})\sqrt{p_*(\boldsymbol{y})} \cdot \nabla\frac{p_t(\boldsymbol{y})}{p_*(\boldsymbol{y})} - \sqrt{p_*(\boldsymbol{x})} \cdot \nabla\frac{p_t(\boldsymbol{x})}{p_*(\boldsymbol{x})}\right\|^2 \mathrm{d}\boldsymbol{x} \\
\leq &\beta \cdot \mathrm{KAE}
\end{aligned} \tag{67}$$

where the last inequality follows from the $p_t$ warm assumption. Suppose we control the KAE by Eq. 64, which means $\partial_t D_\chi(p_t, p_*) \leq 4\beta\epsilon$, and time $T = -C\ln\epsilon$ which leads $p_t$ to the target region, i.e., $\mathrm{KL}(p_t\|p_*) \leq \epsilon$ by the linear convergence, when $\epsilon$ is small enough, e.g., $\epsilon \leq (16\beta C)^{-2}$ and $D_\chi(p_0, p_*) \leq 1/4$, we have

$$D_\chi(p_T, p_*) \leq D_\chi(p_0, p_*) + 4\beta\epsilon T = D_\chi(p_0, p_*) - 4C\beta\epsilon\ln\epsilon \leq D_\chi(p_0, p_*) + 4C\beta\sqrt{\epsilon} \leq 1/2.$$

Hence, the proof is completed. $\qquad\square$

With these Lemmas, we provide the main theorem proof in the following.

*Proof.* Suppose $H_{p_*}(p_t) := \mathrm{KL}(p_t\|p_*)$ for abbreviation. According to the time derivative of KL divergence along any flow, we have

$$\frac{\mathrm{d}}{\mathrm{d}t}H_{p_*}(p_t) = \int_{\mathbb{R}^d} \frac{\delta H_{p_*}}{\delta p}(p_t)\,\partial_t p_t \mathrm{d}\boldsymbol{x}. \tag{68}$$

Therefore, along Remark 2, we have

$$\begin{aligned}
\frac{\mathrm{d}}{\mathrm{d}t}H_{p_*}(p_t) &= \int_{\mathbb{R}^d} \nabla\ln\frac{p_t(\boldsymbol{x})}{p_*(\boldsymbol{x})} \cdot \phi_t(\boldsymbol{x})p_t(\boldsymbol{x})\mathrm{d}\boldsymbol{x} \\
&= -\int_{\mathbb{R}^d}\int_{\mathbb{R}^d} k(\boldsymbol{x}, \boldsymbol{y})p_t(\boldsymbol{x})p_t(\boldsymbol{y}) \cdot \left[\nabla\ln\frac{p_t(\boldsymbol{x})}{p_*(\boldsymbol{x})} \cdot \nabla\ln\frac{p_t(\boldsymbol{y})}{p_*(\boldsymbol{y})}\right]\mathrm{d}\boldsymbol{y}\mathrm{d}\boldsymbol{x},
\end{aligned} \tag{69}$$

which follows from Eq. 36. By taking

$$k(\boldsymbol{x}, \boldsymbol{y}) = (p_*(\boldsymbol{x}))^{-1/2} \, \overline{k}_\sigma(\boldsymbol{x}, \boldsymbol{y}) \, (p_*(\boldsymbol{y}))^{-1/2} \, ,$$

Eq. 69 satisfies

$$
\begin{aligned}
\frac{\mathrm{d}}{\mathrm{d}t} H_{p_*}(p_t) = & - \int_{\mathbb{R}^d} \frac{p_t(\boldsymbol{x})}{\sqrt{p_*(\boldsymbol{x})}} \nabla \ln \frac{p_t(\boldsymbol{x})}{p_*(\boldsymbol{x})} \mathrm{d}\boldsymbol{x} \cdot \int_{\mathbb{R}^d} \overline{k}_\sigma(\boldsymbol{x}, \boldsymbol{y}) \frac{p_t(\boldsymbol{y})}{\sqrt{p_*(\boldsymbol{y})}} \nabla \ln \frac{p_t(\boldsymbol{y})}{p_*(\boldsymbol{y})} \mathrm{d}\boldsymbol{y} \\
= & - \int_{\mathbb{R}^d} \frac{p_t(\boldsymbol{x})}{\sqrt{p_*(\boldsymbol{x})}} \nabla \ln \frac{p_t(\boldsymbol{x})}{p_*(\boldsymbol{x})} \cdot \left[ \int_{\mathbb{R}^d} \overline{k}_\sigma(\boldsymbol{x}, \boldsymbol{y}) \frac{p_t(\boldsymbol{y})}{\sqrt{p_*(\boldsymbol{y})}} \nabla \ln \frac{p_t(\boldsymbol{y})}{p_*(\boldsymbol{y})} \mathrm{d}\boldsymbol{y} \right. \\
& \left. - \frac{p_t(\boldsymbol{x})}{\sqrt{p_*(\boldsymbol{x})}} \nabla \ln \frac{p_t(\boldsymbol{x})}{p_*(\boldsymbol{x})} + \frac{p_t(\boldsymbol{x})}{\sqrt{p_*(\boldsymbol{x})}} \nabla \ln \frac{p_t(\boldsymbol{x})}{p_*(\boldsymbol{x})} \right] \mathrm{d}\boldsymbol{x} \\
= & - \int_{\mathbb{R}^d} p_*(\boldsymbol{x}) \cdot \left\| \frac{p_t(\boldsymbol{x})}{p_*(\boldsymbol{x})} \nabla \ln \frac{p_t(\boldsymbol{x})}{p_*(\boldsymbol{x})} \right\|^2 \mathrm{d}\boldsymbol{x} \\
& - \int_{\mathbb{R}^d} \frac{p_t(\boldsymbol{x})}{\sqrt{p_*(\boldsymbol{x})}} \nabla \ln \frac{p_t(\boldsymbol{x})}{p_*(\boldsymbol{x})} \cdot \left[ \int_{\mathbb{R}^d} \overline{k}_\sigma(\boldsymbol{x}, \boldsymbol{y}) \frac{p_t(\boldsymbol{y})}{\sqrt{p_*(\boldsymbol{y})}} \nabla \ln \frac{p_t(\boldsymbol{y})}{p_*(\boldsymbol{y})} \mathrm{d}\boldsymbol{y} \right. \\
& \left. - \frac{p_t(\boldsymbol{x})}{\sqrt{p_*(\boldsymbol{x})}} \nabla \ln \frac{p_t(\boldsymbol{x})}{p_*(\boldsymbol{x})} \right] \mathrm{d}\boldsymbol{x}. \\
\leq & - \frac{1}{2} \int_{\mathbb{R}^d} p_*(\boldsymbol{x}) \cdot \left\| \frac{p_t(\boldsymbol{x})}{p_*(\boldsymbol{x})} \nabla \ln \frac{p_t(\boldsymbol{x})}{p_*(\boldsymbol{x})} \right\|^2 \mathrm{d}\boldsymbol{x} \\
& + \frac{1}{2} \underbrace{\int_{\mathbb{R}^d} \left\| \frac{p_t(\boldsymbol{x})}{\sqrt{p_*(\boldsymbol{x})}} \nabla \ln \frac{p_t(\boldsymbol{x})}{p_*(\boldsymbol{x})} - \int_{\mathbb{R}^d} \overline{k}_\sigma(\boldsymbol{x}, \boldsymbol{y}) \frac{p_t(\boldsymbol{y})}{\sqrt{p_*(\boldsymbol{y})}} \nabla \ln \frac{p_t(\boldsymbol{y})}{p_*(\boldsymbol{y})} \mathrm{d}\boldsymbol{y} \right\|^2 \mathrm{d}\boldsymbol{x}}_{\text{Kernel Approximation Error(KAE)}} .
\end{aligned}
\tag{70}
$$

the decreasing of KL divergence of SVGD at time $t$ satisfies

$$
\begin{aligned}
\frac{\mathrm{d}}{\mathrm{d}t} H_{p_*}(p_t) & \leq - \frac{1}{4} \int_{\mathbb{R}^d} p_*(\boldsymbol{x}) \cdot \left\| \frac{p_t(\boldsymbol{x})}{p_*(\boldsymbol{x})} \nabla \ln \frac{p_t(\boldsymbol{x})}{p_*(\boldsymbol{x})} \right\|^2 \mathrm{d}\boldsymbol{x} + 4\epsilon \\
& \leq - \frac{\mu}{16} H_{p_*}(p_t) + 4\epsilon,
\end{aligned}
\tag{71}
$$

where the first inequality follows from Lemma 4.3 and the second one follows from Lemma 4.2. However, Lemma 4.2 requires a local condition of $p_t$ which we proved in Lemma C.1. By applying Gronwall's lemma, Eq. 71 implies the desired bound

$$H_{p_*}(p_t) \leq \max \left( 0, \left( H_{p_*}(p_0) - \frac{64\epsilon}{\mu} \right) \right) \cdot \exp \left( -\frac{\mu t}{16} \right) + \frac{64\epsilon}{\mu}. \tag{72}$$

Hence, the proof is completed. □

**Remark 3.** *Actually, instead of the constant upper bound of the density ratio provided in Assumption [$A_3$], we allow the upper bound of the density ratio to be upper-bounded as*

$$\sup_{\boldsymbol{x} \in \mathbb{R}^d} p_t(\boldsymbol{x})/p_*(\boldsymbol{x}) \leq P(t).$$

*where $P(t)$ denotes a polynomial function. Without loss of generality, we suppose $P(t) \leq (t+1)^q$. In this condition, reweighted SVGD can achieve an $O(1/\epsilon)$ when we choose*

$$\sigma_t = \min \left( 1, e^{-t-1} \right). \tag{73}$$

*In the following, we will show how this choice affects the kernel approximation error shown in Lemma 4.3. Similar to Eq. 63, we can obtain the following inequality*

$$
\begin{aligned}
\text{Term 1} \leq{} & 12LM\sqrt{P(t)}\sigma_t \cdot \left(16Cd + \frac{9P(t)Cd}{2M} + 6\sqrt{P(t)}L + 3CL + P(t)CL\right) \\
\leq{} & 192LMCd \cdot \sqrt{P(t)}\sigma_t + 54LCd \cdot P(t)^{1.5}\sigma_t \\
& + 36L^2M \cdot \sqrt{P(t)}\sigma_t + 12L^2MC \cdot P(t)^{1.5}\sigma_t.
\end{aligned}
\tag{74}
$$

*We will easily obtain that*

$$
P^{1.5}(t)\sigma_t = (1+t)^{1.5q} \cdot e^{-(t+1)} \leq \frac{(1+1.5q)^{1.5q} \cdot e^{-1-1.5q}}{t+1},
$$

*and*

$$
\text{Term 1} \leq \left(192LMCd + 54LCd + 36L^2M + 12L^2MC\right) \cdot (1+1.5q)^{1.5q} \cdot e^{-1-1.5q} \cdot (t+1)^{-1}.
$$

*For abbreviation, we suppose*

$$
C_* = \left(192LMCd + 54LCd + 36L^2M + 12L^2MC\right) \cdot (1+1.5q)^{1.5q} \cdot e^{-1-1.5q}.
$$

*Then, Similar to Eq. 71, we have*

$$
\begin{aligned}
\frac{\mathrm{d}}{\mathrm{d}t}H_{p_*}(p_t) \leq{} & -\frac{1}{4}\int_{\mathbb{R}^d} p_*(\boldsymbol{x}) \cdot \left\|\frac{p_t(\boldsymbol{x})}{p_*(\boldsymbol{x})}\nabla \ln\frac{p_t(\boldsymbol{x})}{p_*(\boldsymbol{x})}\right\|^2 \mathrm{d}\boldsymbol{x} + \frac{4C_*}{1+t} \\
\leq{} & -\frac{\mu}{16}H_{p_*}(p_t) + \frac{4C_*}{1+t}.
\end{aligned}
\tag{75}
$$

*It means the KL divergence $H_{p_*}(p_t)$ satisfies*

$$
H_{p_*}(p_t) \leq 4C_* \cdot \exp\left(-\frac{\mu(t+1)}{16}\right) \cdot \text{Ei}\left(\frac{\mu(t+1)}{16}\right) + H_{p_*}(p_0) \cdot \exp\left(\frac{-\mu t}{16}\right)
\tag{76}
$$

*where $\text{Ei}$ is denoted as*

$$
\text{Ei}(x) = -\int_{-x}^{\infty}\frac{\exp(-t)}{t}\mathrm{d}t.
$$

*According to Abramowitz (1972), we have*

$$
e^{-x}\text{Ei}(x) \leq -\frac{1}{2}\ln(1 - \frac{2}{x})
$$

$$
4C_* \cdot \exp\left(-\frac{\mu(t+1)}{16}\right) \cdot \text{Ei}\left(\frac{\mu(t+1)}{16}\right) \leq -2C_*\ln(1 - \frac{32}{\mu(t+1)}) \leq \frac{64C_*}{\mu(t+1) - 32},
$$

*where the last inequality follows from $\ln(x) \geq 1 - 1/x$. Hence, by requiring RHS of Eq. 76 to be smaller than $\epsilon$, we have $t \geq 64C_*/(\mu\epsilon) + 32/\mu$.*

In the following, we will show that a convergence rate can be obtained by approximating an unknown normalizing constant of the target distribution $p_*$ (similar to Theorem 3.1).

**Proposition C.2.** *Suppose Assumption [$A_1$]-[$A_3$] are satisfied, chi-square $D_\chi(p_0, p_*) \leq 1/4$ and PDF of target distribution $p_*(\boldsymbol{x})$ can be estimated by $\hat{p}_*(\boldsymbol{x})$ satisfying*

$$
p_*(\boldsymbol{x}) = \frac{e^{-f_*(\boldsymbol{x})}}{C_*}, \quad \hat{p}_*(\boldsymbol{x}) = \frac{e^{-f_*(\boldsymbol{x})}}{\hat{C}} \quad \text{where} \quad 0 < C_*, \hat{C} < \infty.
$$

*For any $\epsilon > 0$, if we set reweighted kernel $k$:*

$$
\hat{k}(\boldsymbol{x}, \boldsymbol{y}) = (\hat{p}_*(\boldsymbol{x}))^{-1/2}\,\overline{k}_\sigma(\boldsymbol{x}, \boldsymbol{y})\,(\hat{p}_*(\boldsymbol{y}))^{-1/2}, \quad \overline{k}_\sigma(\boldsymbol{x}, \boldsymbol{y}) = \tilde{k}_\sigma(\boldsymbol{x} - \boldsymbol{y}) = \sigma^{-d}\tilde{k}(\sigma^{-1}(\boldsymbol{x} - \boldsymbol{y})),
$$

$$
\int_{\mathbb{R}^d}\|\boldsymbol{y}\|^4 \cdot \tilde{k}(\boldsymbol{y})\mathrm{d}\boldsymbol{y} \leq M, \quad \text{and} \quad \left|1 - \int_{\mathbb{R}^d}\overline{k}_\sigma(\boldsymbol{x}, \boldsymbol{x} - \boldsymbol{y})\mathrm{d}\boldsymbol{y}\right| \leq \frac{1}{2\sqrt{2}},
\tag{77}
$$

*where*

$$\sigma = \min\left(1, \frac{\epsilon}{12LM\sqrt{\beta}} \cdot \left(16Cd + \frac{9\beta Cd}{2M} + 6\sqrt{\beta}L + 3CL + \beta CL\right)^{-1}\right), \qquad (78)$$

*then the KL divergence between $p_t$ and $p_*$ satisfies*

$$H_{p_*}(p_t) \leq \max\left(0, \left(H_{p_*}(p_0) - \frac{64\epsilon}{\mu}\right)\right) \cdot \exp\left(-\frac{\mu t}{16} \cdot \frac{\hat{C}}{C_*}\right) + \frac{64\epsilon}{\mu}. \qquad (79)$$

*Proof.* We suppose the following reweighted kernels

$$k(\boldsymbol{x}, \boldsymbol{y}) = (p_*(\boldsymbol{x}))^{-1/2} \, \overline{k}_\sigma(\boldsymbol{x}, \boldsymbol{y}) \, (p_*(\boldsymbol{y}))^{-1/2}$$
$$\hat{k}(\boldsymbol{x}, \boldsymbol{y}) = (\hat{p}_*(\boldsymbol{x}))^{-1/2} \, \overline{k}_\sigma(\boldsymbol{x}, \boldsymbol{y}) \, (\hat{p}_*(\boldsymbol{y}))^{-1/2}$$

lead to different update rules which are

$$\phi_t(\boldsymbol{x}) = \int \left(k(\boldsymbol{x}, \boldsymbol{y}) \nabla \ln p_*(\boldsymbol{y}) + \nabla_{\boldsymbol{y}} k(\boldsymbol{x}, \boldsymbol{y})\right) \mathrm{d}p_t(\boldsymbol{y})$$

$$= -\int k(\boldsymbol{x}, \boldsymbol{y}) \cdot \nabla \ln \frac{p_t(\boldsymbol{y})}{p_*(\boldsymbol{y})} \mathrm{d}p_t(\boldsymbol{y})$$

$$\hat{\phi}_t(\boldsymbol{x}) = \int \left(\hat{k}(\boldsymbol{x}, \boldsymbol{y}) \nabla \ln \hat{p}_*(\boldsymbol{y}) + \nabla_{\boldsymbol{y}} \hat{k}(\boldsymbol{x}, \boldsymbol{y})\right) \mathrm{d}p_t(\boldsymbol{y})$$

$$= \int \left(\hat{k}(\boldsymbol{x}, \boldsymbol{y}) \nabla \ln p_*(\boldsymbol{y}) + \nabla_{\boldsymbol{y}} \hat{k}(\boldsymbol{x}, \boldsymbol{y})\right) \mathrm{d}p_t(\boldsymbol{y})$$

$$= -\int \hat{k}(\boldsymbol{x}, \boldsymbol{y}) \cdot \nabla \ln \frac{p_t(\boldsymbol{y})}{p_*(\boldsymbol{y})} \mathrm{d}p_t(\boldsymbol{y})$$

where the second and the last equations follow from integration by part. Hence, the dynamic of KL divergence under $\tilde{\phi}_t$ becomes

$$\frac{\mathrm{d}}{\mathrm{d}t} H_{p_*}(p_t) = \int_{\mathbb{R}^d} \nabla \ln \frac{p_t(\boldsymbol{x})}{p_*(\boldsymbol{x})} \cdot \hat{\phi}_t(\boldsymbol{x}) p_t(\boldsymbol{x}) \mathrm{d}\boldsymbol{x}$$

$$= -\frac{\hat{C}}{C_*} \int_{\mathbb{R}^d} \nabla \ln \frac{p_t(\boldsymbol{x})}{p_*(\boldsymbol{x})} \cdot \phi_t(\boldsymbol{x}) p_t(\boldsymbol{x}) \mathrm{d}\boldsymbol{x} \qquad (80)$$

$$\leq \frac{\hat{C}}{C_*} \cdot \left(-\frac{\mu}{16} H_{p_*}(p_t) + 4\epsilon\right)$$

where the equation follows from the definition of $\hat{k}$, and the inequality follows from Theorem 3.1. By applying Gronwall's lemma, Eq. 80 implies the desired bound

$$H_{p_*}(p_t) \leq \max\left(0, \left(H_{p_*}(p_0) - \frac{64\epsilon}{\mu}\right)\right) \cdot \exp\left(-\frac{\mu t}{16} \cdot \frac{\hat{C}}{C_*}\right) + \frac{64\epsilon}{\mu}. \qquad (81)$$

$$\square$$

This proposition demonstrates that approximating an unknown normalizing constant will not harm the linear convergence rate of SVGD with reweighted kernels, and only provides an additional factor $\hat{C}/C_*$ in total complexity. The very common question is that it seems that the convergence can be arbitrarily fast when the factor $\hat{C}/C_*$ is large enough. Actually, we should notice this convergence rate only establishes in asymptotic analysis, which means the discretization error cannot be controlled without a tiny step size when $\hat{C}/C_*$ is large. That means a large $\hat{C}/C_*$ usually implies a small step size in practice rather than arbitrarily fast convergence.

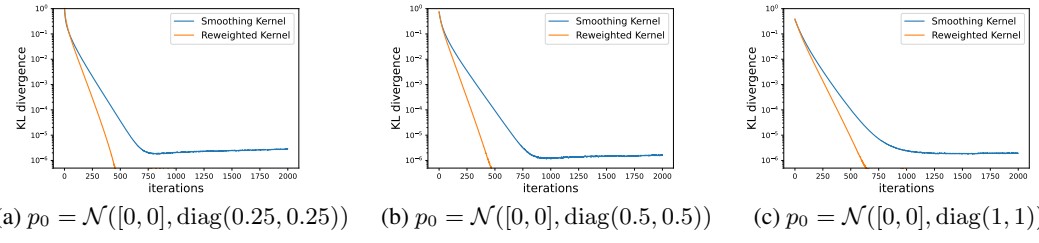

(a) $p_0 = \mathcal{N}([0,0], \mathrm{diag}(0.25, 0.25))$    (b) $p_0 = \mathcal{N}([0,0], \mathrm{diag}(0.5, 0.5))$    (c) $p_0 = \mathcal{N}([0,0], \mathrm{diag}(1, 1))$

Figure 3: Reweighted vs smoothing kernel (1K particles). $p_* = \mathcal{N}([0,0], \mathrm{diag}(5, 1))$.

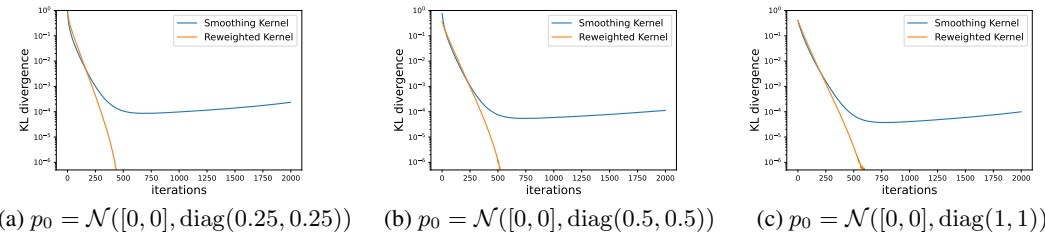

(a) $p_0 = \mathcal{N}([0,0], \mathrm{diag}(0.25, 0.25))$    (b) $p_0 = \mathcal{N}([0,0], \mathrm{diag}(0.5, 0.5))$    (c) $p_0 = \mathcal{N}([0,0], \mathrm{diag}(1, 1))$

Figure 4: Reweighted vs smoothing kernel (0.5K particles). $p_* = \mathcal{N}([0,0], \mathrm{diag}(5, 1))$.

## D  EXPERIMENTAL RESULTS

In this section, we conduct the SVGD with reweighted kernels in some synthetic data to validate our claims, i.e., compared with traditional SVGD, SVGD with reweighted kernels can achieve any $\epsilon$-neighborhood with a linear convergence. To validate our theoretical results in asymptotic settings, we choose different particle sizes and show that sampling by SVGD with reweighted kernels can obtain a lower KL divergence.

To demonstrate the efficiency and stability of SVGD, we provide a numerical illustration for these two algorithms in Figure 5. It is clear that Langevin dynamics suffer from the introduction of the stochasticity, which makes the particle highly unstable. Thus, it is necessary to use more particles to perform the task to guarantee the stability.

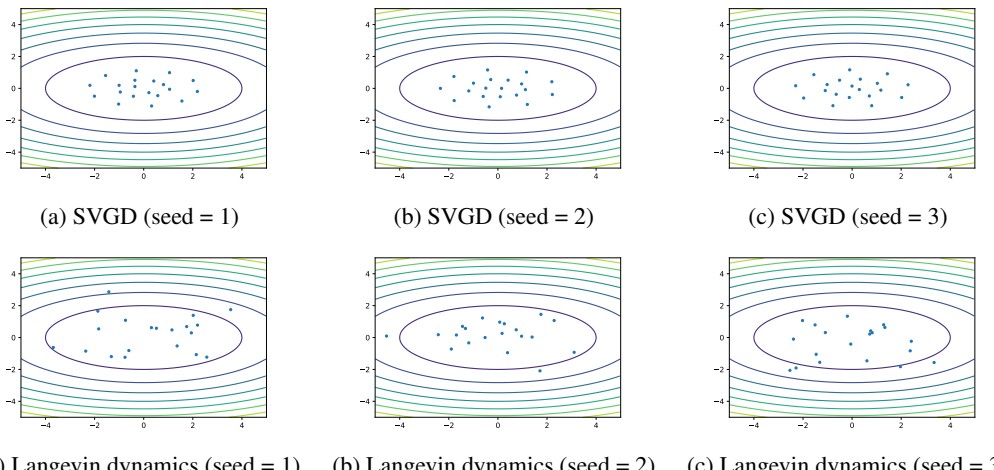

(a) SVGD (seed = 1)      (b) SVGD (seed = 2)      (c) SVGD (seed = 3)

(a) Langevin dynamics (seed = 1)   (b) Langevin dynamics (seed = 2)   (c) Langevin dynamics (seed = 3)

Figure 5: SVGD vs Langevin dynamics.

