# OpenReview forum: "Local KL Convergence Rate for Stein Variational Gradient Descent with Reweighted Kernel"
_ICLR.cc/2023/Conference — Submitted to ICLR 2023_

### Official Review · Reviewer_ecWW · 2022-10-23

**Confidence:** 4
**Correctness:** 4
**Technical Novelty And Significance:** 4
**Empirical Novelty And Significance:** Not applicable
**Recommendation:** 8

**Clarity, Quality, Novelty And Reproducibility:**

The paper is well organized and easy to read. The novelty and significance are sufficient as commented above.

Minor comments:

- Typos in Eq. (10): $\nabla \phi_t (x)$ → $\nabla \phi(x)$ and $\leq S(p_t,q)$ → $\leq 1$ (?)
- The definition of notations $p_{k,z}$ and $p_{*,z}$ in Eq. (16) are missing in the main text.
- What is the constant $C$ in Eq. (21)?

**Strength And Weaknesses:**

**Strengths**:

The convergence analysis of SVGD is challenging because of the vanishing gradient phenomenon caused by the kernel smoothing of the gradient. Previously, Stein LSI was used for the convergence analysis in the related studies, but examples which satisfy this condition are very limited. Instead, this work managed to show the convergence by exploiting the standard LSI  like Langevin dynamics. Moreover, it is also clarified that a reweighted kernel basically provides a sufficient condition for Stein LSI. Thus, this work well contributes to this context.

**Weaknesses**:

- There are several limitations in the main result (Theorem 3.1) compared to Langevin dynamics. For instance, the theory requires a warm start condition $D_X(p_0,p/*)\leq 1/4$ and bounded and smoothness conditions on the trajectory of $\{p_t\}$ in Assumptions (A2) and (A3).  Although the latter conditions are verified empirically in a synthetic experiment, it would be nice if the authors could provide a specific example. Such an example will make the paper stronger.

- A reweighted kernel involves the value of the target distribution $p_*$. Thus, the computation of the normalization term is needed. This computation is not trivial and approximation error to the normalization usually remains. How does the error affect the convergence rate? It would be better to clarify this.

- SVGD with a reweighted kernel using a small $\sigma$ will finally get close to the Wasserstein gradient flow of KL-divergence when $p_t$ is close to $p_*$. Thus, I’m wondering what the advantage is over Langevin dynamics. The empirical or theoretical comparison with Langevin dynamics will be helpful if possible.

**Summary Of The Paper:**

This paper proposes using a reweighted kernel by the target distribution for Stein variational gradient descent (SVGD) and provides its convergence analysis. Specifically, the authors show the convergence of the proposed method under the standard log-Sobolev inequality (LSI) with additional assumptions on the trajectory which are empirically verified by the synthetic experiment.

**Summary Of The Review:**

The paper certainly contributes to the context by giving a specific example for Stein LSI. But there are still limitations compared to Langevin dynamics. Examples that satisfy additional assumptions (A2) and (A3) would increase the significance of the paper.

-- After reading the author's response --
My concerns have been well addressed. Indeed, I agree with reviewer 8ZrX that assumptions are rather strong. However, I think this paper suggests a new strategy for the convergence analysis of SVRG under Stein LSI, which is a quite challenging problem. Thus, I would like to keep the score.

---

> ### Author Response · Authors · 2022-11-08
> **Thank you for your professional review**
>
> We would like to thank the reviewer for the constructive feedback. Below are comments that may hope to address your concerns.
>
> **Weaknesses:**
>
> **Q1:** There are several limitations in the main result ...
>
> **A1:** Thank you for your suggestion. Unlike variants of the Langevin dynamic, both the transition kernel and the stationary distribution in SVGD remain unknown due to the time-inhomogeneous semigroup, which means we can hardly track the density ratio (A3) and smoothness (A2) as previous work, e.g., Lemma 11.4 in [5], even if the initial distribution is given. It is also why we conduct empirical verifications. Nevertheless, we agree that finding a specific example that satisfied Assumptions (A2) and (A3) is an important question, and we will try to answer it in our future work.
>
> **Q2:** A reweighted kernel involves the value of the target distribution ...
>
> **A2:** Thank you for your attention! We provide an additional proposition (Proposition C.2) in Appendix C in the latest version, which demonstrates that approximating an unknown normalizing constant will not harm the linear convergence rate of SVGD with reweighted kernels, and it only provides an additional factor, related to the exact and approximated normalizing constants, in the total complexity.
>
> **Q3:** SVGD with a reweighted kernel using a small ...
>
> **A3:** Good question. In fact, this question is equivalent to the motivation of SVGD. The key difference between Langevin dynamics and SVGD is the $\nabla\log p_t$ term, where Langevin dynamics leverages Brownian motion and SVGD uses repulsive force. As a result, the particles in SVGD interact with each other. Intuitively, this makes the sample efficiency is higher. The empirical gains from SVGD has been validated by other papers [1,2]. Our paper tend to solve a special issue that plain SVGD cannot converge linearly in theory [3], which can be easily obtained in Langevin dynamics [4]. More empirical investigations from this direction can be interesting future works. We also include an illustration in our Appendix D.
>
> **Q4:** Minor comments.
>
> **A3:**  Thank you! We have fixed them in our current version.
>
>
> [1] Liu, Qiang, and Dilin Wang. "Stein variational gradient descent: A general purpose bayesian inference algorithm." *Advances in neural information processing systems* 29 (2016).
>
> [2] Liu, Yang, et al. "Stein variational policy gradient." *arXiv preprint arXiv:1704.02399* (2017).
>
> [3] Duncan, Andrew, Nikolas Nüsken, and Lukasz Szpruch. "On the geometry of Stein variational gradient descent." *arXiv preprint arXiv:1912.00894* (2019).
>
> [4] Vempala, Santosh, and Andre Wibisono. "Rapid convergence of the unadjusted langevin algorithm: Isoperimetry suffices." *Advances in neural information processing systems* 32 (2019).
>
> [5] Mangoubi, O., & Vishnoi, N. K. (2019, June). Nonconvex sampling with the Metropolis-adjusted Langevin algorithm. In Conference on Learning Theory (pp. 2259-2293). PMLR.

---

### Official Review · Reviewer_sU2Z · 2022-10-24

**Confidence:** 2
**Correctness:** 4
**Technical Novelty And Significance:** 3
**Empirical Novelty And Significance:** 1
**Recommendation:** 6

**Clarity, Quality, Novelty And Reproducibility:**

The paper doesn't provide enough details for reproducing the experiments.

The writing is clear.  The proof of the local linear convergence of KL divergence is novel.

**Strength And Weaknesses:**

**Strength**
- The idea of introducing a reweighted kernel is well-motivated.
- The paper proves the convergence rate of KL divergence in a neighborhood of the target.

**Weakness**
- The reweighting requires the knowledge of $p_*$ or $p_t$. They might not be tractable, e.g. the normalizing constant is unknown. The idea doesn't lead to a practical algorithm.
- The experiments only have toy examples (2d Gaussian with diagonal covariance). SVGD is expected to sample from much more challenging distributions.

**Summary Of The Paper:**

The paper studies the convergence rate of SVGD with a reweighted kernel. The main theoretical result is the local linear convergence rate to the target in the sense of KL divergence using the reweighted kernel.

**Summary Of The Review:**

This paper proves the convergence rate of SVGD with a reweighted kernel. The paper is clearly written. But the empirical results are weak.

---

> ### Author Response · Authors · 2022-11-08
> **Thanks for detailed review**
>
> Thanks for your detailed reading. Below are comments that may hope to address your concerns.
>
> **Weaknesses**
>
> **Q1:** The reweighting requires the knowledge of ...
>
> **A1:** Thank you for your attention! Actually, we are able to obtain $p_t$ in integration by Monte Carlo sampling, which means we do not require to consider the normalizing constant of $p_t$. Besides, we may use some other algorithms to estimate the normalizing constant of $p_*$. In this condition, we provide an additional proposition (Proposition C.2) in Appendix C in the latest version, which demonstrates that approximating an unknown normalizing constant will not harm the linear convergence rate of SVGD with reweighted kernels, and it only provides an additional factor, related to the exact and approximated normalizing constants, in the total complexity.
>
> **Q2:** The experiments only have toy examples ...
>
> **A2:** Thanks. Our goal is not to establish the superiority of SVGD with reweighted kernels by testing it on a battery of challenging high-dimensional instances, but rather to demonstrate that, unlike SVGD with smoothing kernels, SVGD with reweighted kernels has both strong theoretical guarantees (KL convergence) and good numerical performance in our cases. Nevertheless, we think extending our continuous analysis to discrete settings even in real-world scenarios is one of the most important topics in our future works.
>
> **Q3:** The paper doesn't provide enough details for reproducing the experiments ...
>
> **A3:** We will publish our code after maintenance.

---

### Official Review · Reviewer_8ZrX · 2022-10-25

**Confidence:** 4
**Correctness:** 4
**Technical Novelty And Significance:** 2
**Empirical Novelty And Significance:** 2
**Recommendation:** 3

**Clarity, Quality, Novelty And Reproducibility:**

Clarity and Quality:

- I am not convinced by Section 3.1. First, I don't think it is necessary to explain the background from Bayesian ML in this paper. I also disagree with the last sentence because one could have taken the kernel k without reweighting by sigma^d.

- Section 4 is confusing. Here, one would need a more focus writing, with perhaps one idea per subsection. In the current form I don't understand the story line. The relationship between Eq 29, 30, 31 is unclear.

- Around Eq 24: The gradient vanishing problem is explained in a confusing way. Putting the proof of Prop 4.1 in the main text would help the reader, but I am not convinced that this is the bottleneck to KL convergence.

Check the sentences:
"In the population limit, optimal smoothing kernel should be Dirac delta function."
- Why?

"After that, we explain Stein log-Sobolev Inequality (SLSI) the necessary condition for analyzing KL convergence
of SVGD can hardly be verified."
- I don't think that SLSI is necessary

"we provide the convergence analysis for SVGD algorithm in KL divergence."
- Overclaim

Novelty:

The approach of this paper relying on reweighting the kernel for KL convergence is new to my knowledge. The conclusion of the main theorem (Linear KL convergence up to a neighborhood) is good, even if I have concerns about the assumptions and the fact that it would not carry to discrete time.


No reproducibility issue.


MINOR:

Sec 2.2 PL is milder than Strong convexity

Remark after Prop 2.2. Technically SVGD algorithm is not a Monte Carlo estimation of Eq 11. It is an exact implementation of Eq 11, because if p_t is an empirical measure, then an integral wrt to p_t is a finite sum.

Eq 23. Could one directly use a kernel k_t which approximates the reweighted Dirac in order to make the kernel approx error small?









**Strength And Weaknesses:**

Strength:

- I have been wondering for a long time how to use a target distribution specific kernel for SVGD, and the reweighting technique sheds some light.

-Section 1 and 2 are well written and I appreciate the overview of Section 2.

Weaknesses:

- This paper considers the infinite particle regime and the continuous time. Although it sheds light on SVGD algorithm, it is still far from the practice of SVGD (I know that the finite particle regime is an open problem). Besides there is some confusion in the paper about continuous and discrete time. In continuous time, the current analyses of SVGD are rather easy and they do not require the kernel to be bounded. Boundedness of the kernel is used in discrete time to establish the descent lemma (Liu'17, Korba et al' 20 and Salim et al 21). So, in my opinion there is no contradiction between Eq 18 and Eq 19 in continuous time.

- In particular, I am worried about how all these results carry in discrete time, since this is where the kernel usually needs to be smooth.

-Assumptions A2 and A3 are conditions on the trajectory of the dynamics, I don't think that they are more reasonable than a bounded kernel. They would need to be proven. This problem is acknowledged and studied empirically though. But A2 and A3 seem cooked up in order to obtain the final result.

- I would have liked to see a comparison between KSD convergence and KL convergence, and why KSD convergence is weaker (even if I understand the intuition)

-I expect the paper to be difficult to read for non-experts (see below).

**Summary Of The Paper:**

This paper studies the problem of sampling wrt to a distribution p* proportional to exp(-V) given access to the gradient of V. The authors consider the mean field limit of Stein Variational Gradient Descent (SVGD). More precisely, SVGD algorithm is an algorithm that relies on updating sequentially the location of a finite number of particles in order to their empirical distribution to approximate the target distribution p*. To run SVGD algorithm, one has to select a RKHS and use its kernel in the update. In continuous time, and with an infinite number of particles, SVGD algorithm converges to a PDE called the SVGD PDE, or in this paper, just SVGD. This PDE rules the evolution of the distribution p_t of the particles.

SVGD can be seen as a dynamics to minimize the KL divergence w.r.t. p*. One issue with this perspective is that the time derivative of KL along SVGD is equal to the opposite of the squared kernelized gradient of the KL. One would like to obtain the squared gradient of KL in order to use Polyak Lojasiewicz inequality (called Log Sobolev Inequality in this setting). This paper develops a technique to do that.

The main result is that, under LSI and using the reweighted kernel, SVGD PDE converges linearly (in terms of KL) up to a neighborhood.

**Summary Of The Review:**

The paper uses interesting techniques but the exposition is below the standard (especially from Section 3). Besides, the assumptions are too strong, and I wonder if the main message of the paper (linear convergence in KL up to a neighborhood) is actually true.

---

> ### Author Response · Authors · 2022-11-08
> **Thank you for your review (Part 1/2)**
>
> Thanks for your reading. Below are comments that may hope to address your concerns.
>
> **Weaknesses**
>
> **Q1:** This paper considers the infinite particle regime...
>
> **A1:** Sorry for the confusion. Actually, in Section 3.2, we hope to explain the limitations of Stein Log-Sobolev inequality in continuous SVGD analysis. Technically, assumptions in continuous SVGD analysis should be compatible with its discrete version. However, SLSI assumption obviously contradicts kernel boundedness in discrete settings, and the most commonly used smoothing kernels (the RBF kernel) in SVGD are indeed bounded. Besides, it is obscure when SLSI holds.  Therefore,  we introduce reweighted kernels and investigate some other tractable assumptions for providing the KL convergence to SVGD in continuous analysis. It should be noted that the compatibility between Eq. 18 and 19 in continuous settings cannot justify the soundness of SLSI.
>
> **Q2:** In particular, I am worried about how all these results ...
>
> **A2:** In this paper, we mainly focus on the KL convergence rate analysis of SVGD in continuous time. Besides, we also showed some potential to extend these results to a discrete version by Appendix D. There will definitely be some gaps in extending these theoretical results, and it is also one of the most important topics in our future works.
>
> **Q3:** Assumptions A2 and A3 are conditions on the trajectory of the dynamics ...
>
> **A3:** Sorry for the confusion. It should be noted that we do not expect to replace bounded kernel assumption with Assumptions A2 and A3. Instead, compared with the SLSI (contradicting bounded kernels), we expect to investigate the dynamic of KL divergence under Assumptions A2 and A3 because they are more intuitive, verifiable, and compatible with our reweighted kernels.
>
> **Q4:** I would have liked to see a comparison between KSD convergence and KL convergence ...
>
> **A4:** First, we do not claim KSD is weaker than KL, and we focus on KL convergence because the theoretical guarantee of several downstream tasks requires the convergence of KL or Wasserstein 2 distance shown in Section 3.1. Besides, the W2d convergence can be directly derived from KL convergence instead of KSD, except when the RKHS associated with KSD is large enough, and the KL functional properties are good enough. Hence, we think the analysis of KL convergence will benefit our understanding of downstream tasks, and it is also the motivation to replace the limited assumption, i.e., Stein log-Sobolev inequality.
>
> **Clarity and Quality**
>
> **Q1:** I am not convinced by Section 3.1. ...
>
> **A1:** Actually, in Section 3.1, we introduce some downstream tasks to demonstrate that, compared with KSD convergence, there are some important downstream applications requiring the KL convergence of sampling algorithms, which is also the reason why we expect to replace the Stein log-Sobolev inequality. About the $sigma^d$ term, since it is a constant, so it is not quite important when $\sigma$ is Fixed, which only has impact on the gradient scaling. However, if you take different $\sigma$ choices without the normalizing constant, the scaling will explode. For SVGD, when the normalizing constant does not exist, the gradient of any input $x$ will collapse to $\int \nabla \log p_*(x) - \nabla \log p_t(x) dx$, which is usually not integrable and unreasonable (since all gradient are the same). The normalizing constant is widely adapted in smoothing techniques [1,2,3].
>
> **Q2:** Section 4 is confusing. ...
>
> **A2:** Sorry for the confusion, we may explain the story here.  In Section 4.1, we explain why and how we should introduce reweighted kernels to replace traditional smoothing kernels. Then, Section 4.2 provides the dynamic of KL under reweighted kernels. As for Eq 29, 30, 31, they show the derivative of KL divergence can be upper bounded by the functional values under the combination of reweighted kernels and Assumption 2, 3. Such an inequality provides a linear convergence rate of KL and plays a similar role as Stein log-Sobolev in the previous analysis of continuous SVGD. Hence, we called Eq.31 a local "Stein log-Sobolev Inequality" near the target distribution.
>
> **Q3:** Around Eq 24: The gradient vanishing problem is explained in a confusing way. ...
>
> **A3:** Thanks for the suggestion. We further explain why the gradient vanishing problem will be the bottleneck to KL linear convergence in our latest version. Actually, KL convergence of SVGD has been proven in [4], and it should be noted that the gradient vanishing problem is the bottleneck of the linear convergence rate of KL. Because the biased Wasserstein gradient estimation in smoothing kernel SVGD will have an additional $p_t$ term, which hampers the sufficient descent of KL divergence at each iteration and even the convergence rate. Therefore, we have shown that with proper modifications, linear convergence can indeed be obtained.

---

> > ### Comment · Reviewer_8ZrX · 2022-11-14
> > **Thanks for the answer**
> >
> > I appreciate the answer but I still believe that the assumptions (A2 and A3) are too strong to study a continuous time version of SVGD in the population limit. If everything was discretized, maybe, because one has to deal with several sources of error.
> >
> > If the analysis relies on stringent assumptions already in continuous time, I don't think that the analysis could be applied to the SVGD algorithm, which is the final goal.
> >
> >
> > EDIT: More details here:
> >
> > **Weaknesses:**
> >
> > A1 and A4. OK. I understand that the authors want to find a replacement for SLSI to obtain linear KL convergence. It is somehow written in the paper, but could be further clarified.
> >
> > A2 and A3. Thanks for pointing to Appendix D. To be clear, I think that the ICLR community is more interested in the SVGD algorithm (discrete space, discrete time) than the SVGD PDE (continuous space, continuous time). Results on SVGD PDE are more suitable in a dynamical system venue in my opinion.
> >
> >
> > In general, obtaining results for the algorithm is much more difficult than for the PDE (for example, the analysis of the SVGD algorithm was an open problem until two weeks ago). So, if this paper studies the SVGD PDE, then the results obtained here should be clean/simple enough to be transferrable to the SVGD algorithm.
> >
> > However, this paper, which is about continuous space and time, relies on some heavy analysis, see Appendix C or **assumptions on the trajectory of the algorithm (A2 and A3), see main text**. These assumptions were not proven but only checked through experiments. Although I appreciate the effort made by the authors to justify these assumptions through experiments, I have doubt that the results of this paper could be carried out in discrete space and time.
> >
> > **Clarity:**
> >
> > A1. Yes, there is some intuition for the scaling $\sigma^d$, we somehow would like the kernel to be the density of the Gaussian distribution and that's why we can normalize. But at the end of the day the scaling is arbitrary, because the kernel doesn't need to normalize to one. And, I can use the scaling to drive the gradient to zero (vanishing gradient problem) or the make the gradient explode. To summarize, I am saying that the scaling does not explain the vanishing gradient problem, because the scaling is arbitrary.
> >
> > That being said, I thought that if $k_\sigma$ is the normalized Gaussian kernel, $$E_{y \sim p_t} \left(k_\sigma(x,y)\nabla \log \frac{p_t}{p_*}(y)\right) = \int d p_t(y) k_\sigma(x,y) \nabla \log \frac{p_t}{p_*}(y) = E_{y \sim N(x,\sigma^2)} \left(p_t(y) \nabla \log \frac{p_t}{p_*}(y)\right),$$
> > converges to $p_t(x) \nabla \log \frac{p_t}{p_*}(x) \neq 0$ as $\sigma \to 0$, as explained in blue on Page 7.
> >
> > In general, I think that the last paragraph of Section 3.1 needs some clarifications.
> >
> > A2 and A3. Thanks for the clarification
> >
> > A4. Very interesting comment, thanks!
> >
> > A5. Do you mean that linear convergence of KL implies SLSI? I don't think that Korba et al are saying that, are they?
> >
> > Thanks for the answers to the minor points.
> >
> >
> > *Conclusion: The idea of reweighting the kernel to obtain KL rates makes sense, but the theory has some gaps.*

---

> > ### Author Response · Authors · 2022-11-21
> > **Response to Reviewer 8ZrX**
> >
> > Thanks for your response. Notably, the assumptions are not required to be so stringent, which has been explained in our Remark 1. In the latest revision, we provide Remark 3 in Appendix C to show that we allow the density ratio to grow with $t$ with a polynomial. While in the main text, we choose the mild assumptions for intuitive proof to clarify our results. Besides, we have provided the motivation and empirical validation for A2 and A3 in Sec. 3.2.
> >
> > More importantly, we have to emphasize that our main contribution is to show the effectiveness and efficiency of SVGD in a standard metric, KL divergence (rather than KSD as previous analysis), which is concerned with many downstream tasks. Moreover, it is also the intuition to remove the limited assumption SLSI and investigate new tractable and relatively mild assumptions. We acknowledge that extending the analysis to the discrete version is very important. However, the current contraction between SLSI and bounded kernel makes it hopeless to provide KL convergence of SVGD, and we believe we have provided some new potentials.

---

> ### Author Response · Authors · 2022-11-08
> **Thank you for your review (Part 2/2)**
>
> **Q4:** Why? : Check the sentences: "In the population limit, optimal smoothing kernel should be Dirac delta function."
>
> **A4:** Smoothing kernel is designed for particle-based settings, which performs a bias-variance tradeoff to estimate the population dependent parameters / functions (such as the Wasserstein gradient here). Larger $\sigma$ in smoothing kernel leads to less variance (from empirical particles) but larger bias. However, in the population limit, there isn’t any empirical randomness from particles, thus we can choose $\sigma=0$ to perform the unbiased estimation without any empirical variance.
>
> **Q5:** I don't think that SLSI is necessary ...
>
> **A5:** Although the convergence of KL can be proved [4], the convergence rate in KL divergence cannot be obtained without SLSI [5].
>
> **Q6:** Overclaim: "we provide the convergence analysis for SVGD algorithm in KL divergence."
>
> **A6:** Thank you for your reminder. We have revised it in the latest version.
>
> **MINOR**
>
> **Q1:** Sec 2.2 PL is milder than Strong convexity
>
> **A1:** Thank you for your attention. We have revised it in the latest version.
>
> **Q2:** Eq 23. Could one directly use a kernel k_t which approximates ...
>
> **A2:** We are not quite sure what is k_t refered in the review. We suppose k_t means $k(x,y) = \left(p_t(x)\right)^{-1/2}\overline{k}_\sigma(x,y)\left(p_t(y)\right)^{-1/2}$. In this condition, SVGD algorithm requires $\nabla p_t$ which can hardly be estimated.
>
> [1] Witkin, Andrew P. "Scale-space filtering." *Readings in Computer Vision*. Morgan Kaufmann, 1987. 329-332.
>
> [2] Altman, Naomi S. "Kernel smoothing of data with correlated errors." *Journal of the American Statistical Association*
>  85.411 (1990): 749-759.
>
> [3] Wand, Matt P., and M. Chris Jones. *Kernel smoothing*. CRC press, 1994.
>
> [4] Liu Q. Stein variational gradient descent as gradient flow[J]. Advances in neural information processing systems, 2017, 30.
>
> [5] Korba, A., Salim, A., Arbel, M., Luise, G., & Gretton, A. (2020). A non-asymptotic analysis for Stein variational gradient descent. Advances in Neural Information Processing Systems, 33, 4672-4682.

---

### Decision · Program_Chairs · 2023-01-20

**Decision:**

Reject

**Justification For Why Not Higher Score:**

Work is mainly theoretical with quite a few gaps that need to be addressed before it leads to a practical method.

**Justification For Why Not Lower Score:**

N/A

**Metareview: Summary, Strengths And Weaknesses:**

First, I want to thank the authors for engaging with the reviewers and providing detailed comments.
There was a strong disagreement between two reviewers, both with a (warranted) high confidence score. They both appreciated the use of the reweighting trick but one reviewer was concerned that the extension to discrete time was not trivial and that the analysis in continuous time was not suitable to ICLR.

After careful consideration and looking at the paper, I ultimately decided to mainly agree with reviewer 8ZrX. Not that the current work is not enough in itself, as reviewer ecWW pointed out, but rather that in its current theoretical form, the paper is not the best match for the ICLR audience. I understand the appeal of such a large conference but I reckon this work would have more impact in a more theoretically oriented venue like COLT or even ICML.